# Precise visuomotor transformations underlying collective behavior in larval zebrafish

Roy Harpaz [1,2 ✉], Minh Nguyet Nguyen[3], Armin Bahl[4] & Florian Engert [1,2]

Complex schooling behaviors result from local interactions among individuals. Yet, how sensory signals from neighbors are analyzed in the visuomotor stream of animals is poorly understood. Here, we studied aggregation behavior in larval zebrafish and found that over development larvae transition from overdispersed groups to tight shoals. Using a virtual reality assay, we characterized the algorithms fish use to transform visual inputs from neighbors into movement decisions. We found that young larvae turn away from virtual neighbors by integrating and averaging retina-wide visual occupancy within each eye, and by using a winner-take-all strategy for binocular integration. As fish mature, their responses expand to include attraction to virtual neighbors, which is based on similar algorithms of visual integration. Using model simulations, we show that the observed algorithms accurately predict group structure over development. These findings allow us to make testable predictions regarding the neuronal circuits underlying collective behavior in zebrafish.

[1] Department of Molecular and Cellular Biology, Harvard University, Cambridge 02138, USA. [2] Center for Brain Science, Harvard University, Cambridge, MA 02138, USA. [3] Solomon H. Snyder Department of Neuroscience, Johns Hopkins University, Baltimore, MD 21205, USA. [4] Centre for the Advanced Study of Collective Behaviour, University of Konstanz, Konstanz 78464, Germany. ✉email: harpazone@gmail.com

Complex collective behaviors such as schooling in fish and flocking in birds can result from local interactions between individuals in the group[1–6]. Understanding how sensory signals coming from surrounding neighbors guide fast and accurate movement decisions is therefore central to the understanding of emergent collective behavior and is of great interest from both computational and neurobiological perspectives[7].

Theoretical models treating groups of animals as physical systems, suggested 'simple' interactions among individuals and explored the collective group states that emerge from these interactions[2,3,5,6,8–12]. Experimental studies, relying on recent advances in machine vision and tracking algorithms[13–18], attempted to infer individual interaction rules directly from animal movement trajectories and compared them to the hypothesized rules from theoretical studies[19–26]. Commonly, such interaction rules assume that an individual animal classifies all neighbors as individual objects, and that various computations are subsequently performed on these objects. These computations include estimating every neighbor's distance, orientation or velocity and performing mathematical operations such as averaging and counting on these representations[2–6], or to selectively respond to specific neighbors but not to others[19,20,23]. Alternatively, complex collective behaviors can also emerge from more simplified computations, which rely primarily on the spatial and temporal low-level statistics of retinal inputs[10,27,28]. Specifically, several theoretical models have used the visual projection of neighbors on the retina as the sole input to the animal and explored the resulting collective behavior[10,27,28]. Whether or not animals use representations of their neighbors as individual objects, and perform complex computation on these representations or whether they base their behavioral decisions on global sensory inputs is currently unknown in most animal species. Consequently, the brain mechanisms and neurobiological circuits involved in collective social behavior are mostly unknown as well.

The zebrafish model system is uniquely situated to help address this gap in knowledge. First, this fish species exhibits complex social behaviors, even at the larval stage, which are expected to have a strong visual component ([20,24,26,29–31], but see refs. [32,33] for other modalities). Second, previous studies in larval zebrafish have successfully characterized the underlying computations and brain mechanisms of other complex behaviors such as prey capture[34–38], predator avoidance[39,40], motor learning[41], and decision making[42–44]. In many of these studies, virtual reality (VR) assays were used to systematically probe and analyze the behavioral responses of the fish, and recently, VR assays were also shown to successfully elicit social responses in larval and juvenile zebrafish[31,45,46]. Third, the zebrafish is genetically accessible[47] and, at the larval stage, can be studied using various imaging and neural activity recording techniques[48–52]. Recently, new insights into the molecular pathways involved in social and collective behavior have started to emerge, detecting unique genes and neuropeptides associated with social behavior and the detection of conspecifics[53–56]. Therefore, the larval zebrafish can be used to study the specific visuomotor transformation involved in collective behavior as they emerge during development, and the neurobiological circuits at their basis.

We analyzed here collective swimming behavior in groups of larvae at different developmental stages. We detect complex group structure already at 7 days post fertilization (dpf), which strongly depends on visual information and continues to develop as fish mature. We then utilized a virtual reality assay[31,45,46] to vary the static and dynamic features of naturalistic as well as artificial stimulus patterns, and tested the effects of varying the statistics of these patterns on the movement decisions of the fish. Using this assay, we characterized the precise visuomotor transformations that control individual movement decisions and the interaction rules that allow fish to integrate information from multiple neighbors. Studying these transformations over development allowed us to hypothesize which of these computations are already mature in the younger larvae, and which computations continue to evolve over development. Using model simulations, we verified that the identified visuomotor transformations can accurately account for the observed collective swimming behavior of groups. Finally, we used our findings to formulate predictions about the structure and function of the neural circuits that are involved in transforming visual input into movement decisions.

## Results

**Group structure in larval zebrafish depends on visual social interactions**. To understand how social interactions shape group structure over development, we studied collective swimming behavior in groups of 5 or 10 larval zebrafish at the ages 7, 14, and 21 dpf, swimming freely in closed arenas. (Fig. 1a, Movies 1–3, "Methods"). We find that already at 7 dpf, larval zebrafish respond to their neighbors, with groups exhibiting increased dispersion compared to chance levels (Fig. 1b, c, Fig. S1a, "Methods"). Group structure completely disappeared when fish were tested in total darkness, confirming the strong visual component of the interactions (dispersion$_{7 \text{ dpf, light}}$ = 0.075 ± 0.067, dispersion$_{7 \text{ dpf, dark}}$ = 0.01 ± 0.068 [mean ± SD]) (Fig. 1b, c). As fish matured, this repulsive tendency reversed and fish swam towards their neighbors, resulting in an age-dependent increase in group cohesion (dispersion$_{14 \text{ dpf}}$ = −0.097 ± 0.083, dispersion$_{21 \text{ dpf}}$ = −0.55 ± 0.24 [mean ± SD]), as reported previously[20,30,31] (Fig. 1d). Average swimming speed and alignment between fish also increased over development, while bout rate decreased (Fig. S1b–d). Among these developmental changes in behavior, we focus here on the aggregation behavior of the fish and its unique developmental trajectory.

To understand how a focal fish responds to visual information from its neighbors, we estimated the angular occupancy that neighbors projected onto the two retinae of the focal fish[57,58] (Fig. 1e, Movie 4). We found that even a simplified global statistic of the visual input, such as the difference between total occupancy experienced on each of the retinae, seemed to modulate the observed turning directions of the focal fish (Fig. 1f). Specifically, at 7 dpf, fish turned away from the more occupied eye, and the strength of the turning response steadily increased as the difference in occupancy between the retinae increased. At ages 14 and 21 dpf, on the other hand, fish turned toward the more occupied side, and this response peaked at intermediate values of difference in retinal occupancy, while even larger differences in retinal occupancy led to a decrease of the response (Fig. 1f). No modulation of turning was observed for fish swimming in total darkness (Fig. 1f), in accordance with the lack of group structure in the dark. In addition to turning direction, we observed that the bout rate of the fish was modulated by the total integrated retinal occupancy experienced by the larvae, in which bout rate was maximal for low occupancy values (Fig. S1e).

Together, these results show that visually mediated complex social interactions can be detected already at 7 dpf and that these interactions transition from repulsive to strongly attractive by age 21 dpf. In addition, a simple global statistic representing visual occupancy on the retinae might be sufficient to explain these behaviors. Next, we use a virtual reality assay to explicitly test fish responses to retinal occupancy and to infer the algorithms that allow fish to respond to complex visual scenes with multiple neighbors.

**Virtual reality reveals that young larvae specifically respond to retinal occupancy**. To specifically test fish responses to retinal occupancy and to reveal the algorithms used to integrate

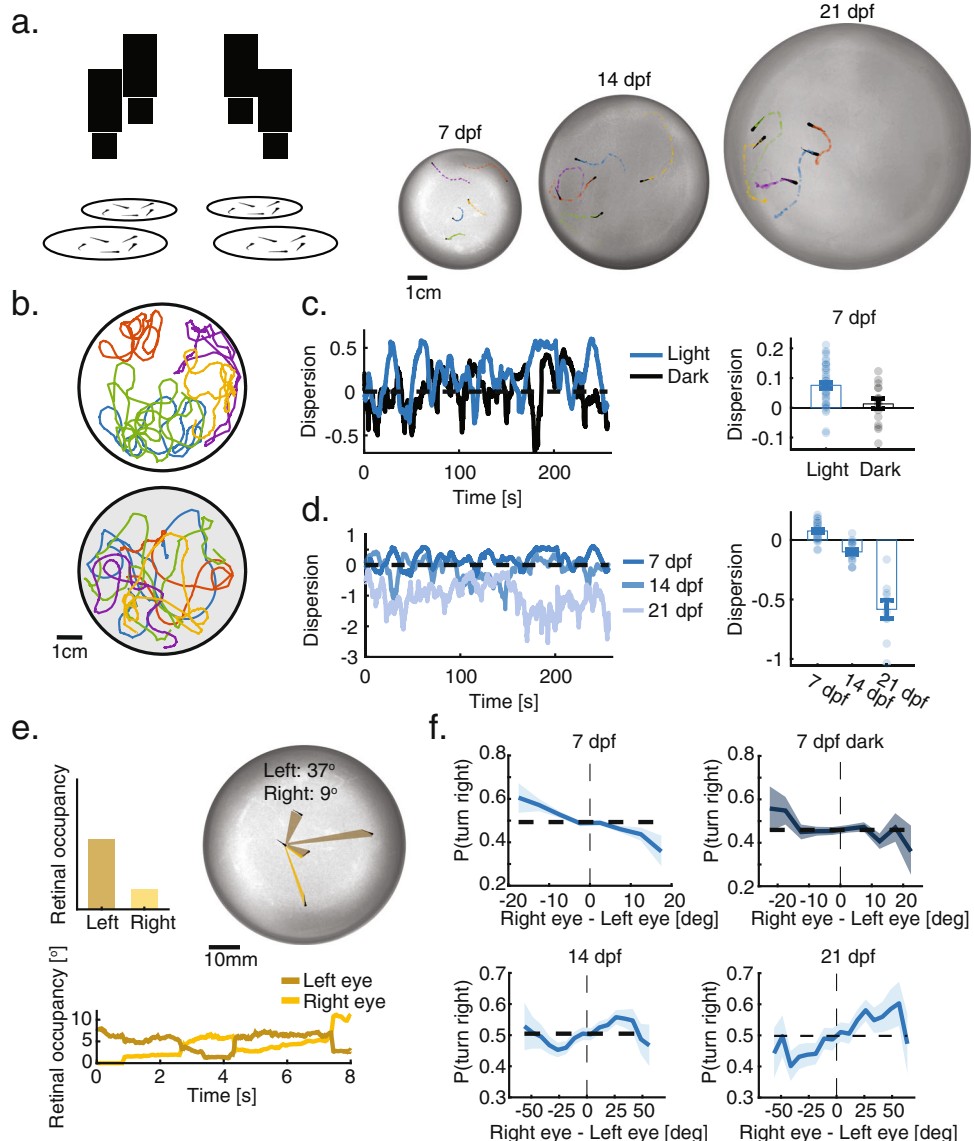

**Fig. 1 Group structure depends on visual interactions and develops with age. a** Left: Experimental system. Multiple cameras capture the behavior of multiple groups and individuals swimming freely in separate arenas. Right: Example images and trajectories of groups of 5 larvae at 7, 14, and 21 dpf. Different colors represent different fish in the groups. **b** Example trajectories (recorded over 5 min) of groups of 5 larvae at age 7 dpf, swimming together in the light (top) or in total darkness (bottom). Different colors represent different fish. **c** Left: Normalized dispersion values for one group swimming together in the light (blue) and in the dark (black). Zero represents the average dispersion value expected when fish do not interact, and positive values represent overdispersed distributions ("Methods"). Right: groups of 7 dpf fish are more dispersed than is expected by chance ($p = 8 \times 10^{-10}$, $N_{light} = 48$ groups, two-sided $t$ test; $t = 7.7$) and are also more dispersed than groups swimming in total darkness ($p = 0.0016$, $N_{dark} = 16$ groups; two sided $t$ test, $t = 3.3$; Cohen's d $= 0.95$). Bars represent mean over groups; error bars are SEM. **d** Left: Example dispersion values for groups at 7, 14, and 21 dpf. Right: At 14 and 21 dpf group are significantly less dispersed (more aggregated) than chance ($p_{14} = 3 \times 10^{-5}$, $p_{21} = 6 \times 10^{-5}$; $N_{14} = 21$, $N_{21} = 10$; two sided $t$ test; $t_{14} = -5.3$, $t_{21} = -7$), and dispersion also decreases significantly over development ($p = 6 \times 10^{-26}$, one-way ANOVA; $F = 137$). Bars represent mean over groups; error bars are SEM. **e** Top: Image showing the total retinal occupancy that neighbors cast on each of the eyes of a focal fish. Bottom: Example traces of the total retinal occupancy for each of the eyes over 8 s. **f** Probability to turn right (per bout) as a function of the difference in retinal occupancy experienced by each eye (negative values - higher occupancy to the left). At 7 dpf, larvae tend to turn away from the more occupied side and do not respond to neighbors in total darkness (Top row). At 14 dpf, fish begin turning towards the more occupied side, and this tendency increases at 21 dpf (Bottom row). Bold lines represent turning probability calculated as the fraction of right turns out of all turns collected from all fish in 5° bins; error bars are the 95% confidence interval of the fitted Binomial distribution to the events in each bin.

information from multiple neighbors we utilized a simplified virtual reality (VR) assay, in which fish respond to projected moving objects around them, mimicking neighboring fish. We begin by focusing on 7 dpf larvae as responses in these fish are expected to be less complex than those observed in older animals (Fig. 1). Previously, older larvae (17–26 dpf) and adults were shown to be attracted to projected moving objects that exhibit movement dynamics of real fish[31,46]. Extending these studies to 7 dpf larvae in our VR assay, we found that fish turn away from projected dots that mimic the motion of real neighbors (Fig. S2a–c, "Methods"), capturing both the group structure and response tendencies observed in our group swimming experiments (Fig. 1c, f).

Next, we varied the physical features, motion dynamics and number of projected objects presented to the fish, to precisely characterize their responses to these features (Fig. 2a, S3a, "Methods"). We generated our stimuli using a pinhole model of the retina of the fish, which transformed bottom-projected stimuli onto retinal space. 'Retinal occupancy' is then defined as the occupied 'pixels' on the retina by the projected stimuli ("Methods") (Fig. 2a, S3b, Movie 5, "Methods"). This model allowed us to independently vary specific features of the stimuli in retinal space while keeping other variables constant.

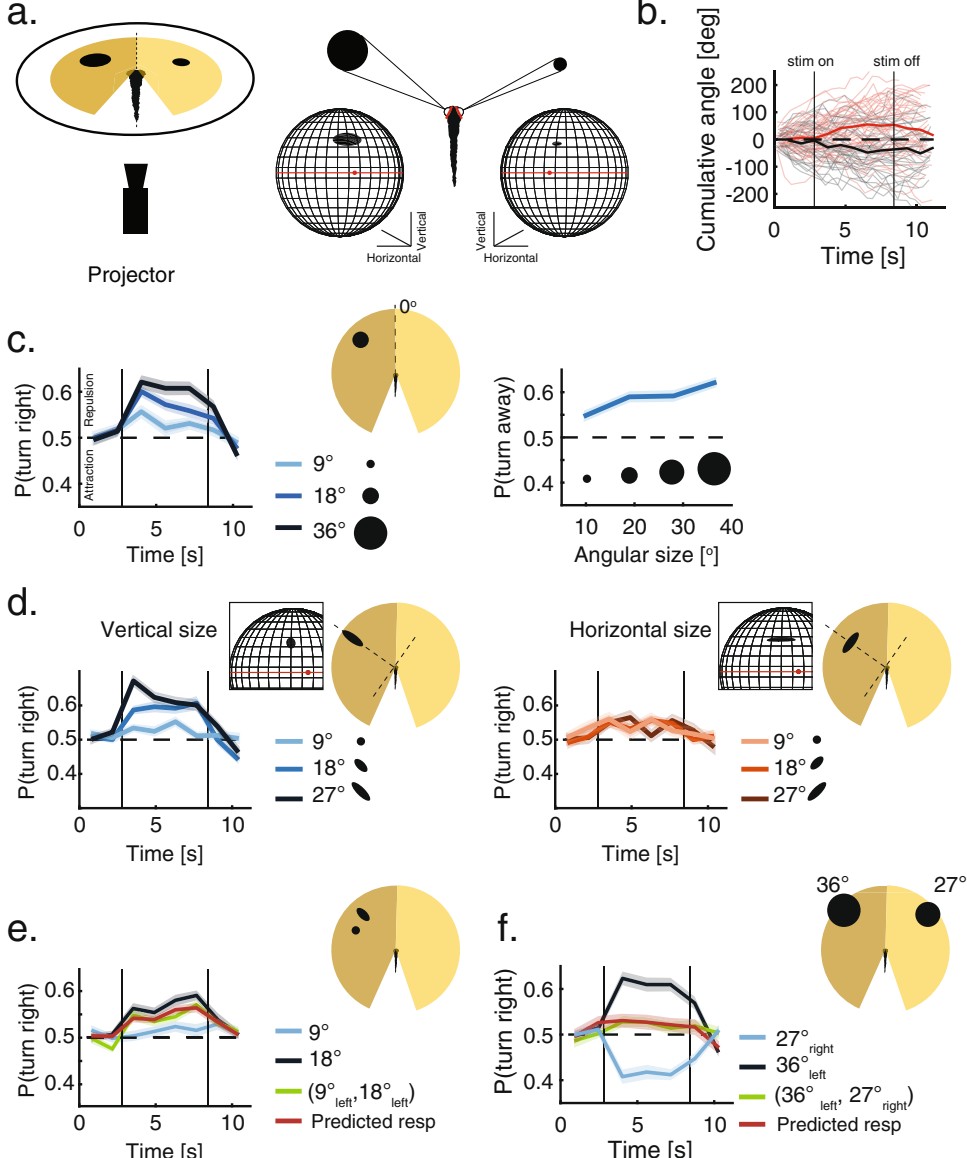

**Fig. 2 Virtual reality reveals the algorithms fish use to integrate visual social information. a** Left: Testing fish social interactions using closed-loop projection of simplified moving objects mimicking neighbors ("Methods"). Right: A pinhole model of the retina is used to estimate the shape, size and position of the image of the projected object on the retina of the fish (black shapes). Both retinae are modeled as spheres, red dots are the center of the back of the retina and red lines represent the horizon line ("Methods"). **b** Examples of the cumulative turning angles for one fish responding to a single dot (36°) mimicking a neighbor moving in bouts in the left visual field (red lines) and in the right one (gray lines) over 40 trials. Bold lines represent averages over trials; vertical lines represent times when stimulus is turned on and off during the trials. **c** Left: Probability to turn right per bout when a single moving dot of different sizes is presented to the left of the fish (N = 24 fish, ""Methods""). Right: Probability to turn away from moving dots of different sizes presented to the left of the fish, calculated over the entire stimulus duration. At 7 dpf, larvae consistently turn away from the side of the projected image. **d** Left: Probability to turn right per bout in response to ellipses of increasing vertical size (perpendicular to the plane of the eye), while the horizontal size remains constant at 9° (N = 32 fish). Inset shows the image of the vertical ellipse on the retina. Right: Probability to turn right per bout in response to ellipses of increasing horizontal sizes (parallel to the plane of the eye), while the vertical size remains constant at 9° (N = 32 fish). Inset shows the image of the horizontal ellipse on the retina. **e** Probability to turn right per bout in response to two images presented together to the left visual field (green line); to each of the images presented alone (blue lines) and the prediction based on the weighted average of the responses to each stimulus presented alone (red line, "Methods"), where weights represent the relative sizes of the stimuli (N = 32 fish). **f** Probability to turn right per bout in response to two dots presented simultaneously to each eye of the fish (green line); to each dot presented alone (blue lines) and the prediction based on the linear summation of the (competing) recorded response biases to each dot presented alone (red line, "Methods")(N = 24 fish). In panels (**c**–**f**), probability to turn right per bout is calculated as the fraction of right turns out of all turns in 1.25 s time bins. Bold lines are average; shaded areas are SEM.

Using this assay, we found that 7 dpf larvae turn away from a moving dot projected either to the left or right visual field. We used dark dots projected on a light background which moved tangentially around the head, from ±60º to ±30º, with intermittent bouts (Fig. 2b, c, Fig. S3a, Movie 6, "Methods"). Increasing the (angular) size of the dot, monotonically increased the probability of the fish to turn away from the stimulus, in agreement with the observed responses to retinal occupancy in group swimming experiments (Fig. 2c, 1f). We also found that responses were qualitatively similar for light dots on a dark background (Fig. S3c), and that repulsion tendencies completely disappeared for small objects occupying <6º on the retina (Fig. S3d). In addition, moving stimuli elicited stronger responses than stationary ones, as expected from simple motion saliency, whereas the explicit motion direction contributed only negligibly to the observed responses (Fig. S3e).

We next tested the effects of different retinal positions by presenting stationary stimuli to different sections of the visual field, while keeping retinal occupancy constant. We found that elevation on the retina (i.e. the radial distance of the project dot) did not modulate the turning responses of the fish (Fig. S3f), while the position in azimuth generated a slight suppression at the edges of the visual field (Fig. S3g).

Importantly, we found that fish repel away from stimuli mostly by modulating the probability of directed turns while keeping other variables such as magnitude of turns (Fig. S3h), the average path traveled in a bout and the overall bout rate constant (Fig. S3i). The lack of modulation of the average path traveled in a bout, indicates that fish responses are consistent with routine turns as opposed to large magnitude escapes[59].

These results confirm the specific role of retinal occupancy in modulating the turning responses of the fish. We next test how visual information is integrated from multiple neighbors and over different dimensions of the retina to guide behavioral decisions.

**Behavioral responses to visual occupancy are based on retina-wide integration and inter-eye competition**. To understand how 7 dpf larvae integrate visual information over the retina we varied the physical dimensions of the projected stimuli and tested fish responses to these changes. We found that stretching the projected dot in the vertical dimension, which changes the height of the image on the retina (Fig. S3b), and increases the magnitude of vertical occupancy specifically, resulted in an increased tendency to avoid the presented stimulus (Fig. 2d, left). Yet, stretching the dot horizontally, thereby changing the width of the image on the retina, and the integrated horizontal occupancy, had no effect on behavior (Fig. 2d, right). The prominent role of the vertical dimension of the stimulus on the retina, was further corroborated by repeating these experiments in bowl-shaped arenas, with stimuli presented to the side of the fish instead of the bottom, which allowed us to stimulate additional positions in retinal space (Fig. S4a). Importantly, we observed similar selectivity to the orientation of the stimuli when multiple identical dots, separated from one another, were arranged vertically (i.e. same angle from the fish, at increasing radial distances) or when they were arranged horizontally (i.e. at different angles from the fish, with the same radial distance): fish increased their tendency to turn away when more dots were presented vertically, yet turned with a similar probability if one, two or three dots were presented horizontally (Fig. S4b).

To further elucidate how visual occupancy is integrated from multiple objects over the retina, we presented to one eye of the fish, two stimuli with different vertical sizes (and similar horizontal sizes). We found that the observed response to the combined presentation of the stimuli was an intermittent value between the two recorded responses to each stimulus presented alone (Fig. 2e). More specifically, the response to the joint presentation of the two stimuli was accurately predicted by a weighted average of the recorded responses to each stimulus presented alone, with weights equal to the relative size of the stimuli (Fig. 2e, S4c, "Methods"). Here again, results were similar regardless of whether the two presented stimuli were clearly separated from one another or if they were joined to create one larger stimulus (Fig. S4d). These results indicate that fish use the different dimensions of the retina differently—they integrate visual occupancy over the vertical dimension of the retina and average the resulting values over the horizontal dimension (see Fig. 4a for illustration). In addition, visual occupancy seems to dominate over the number and density of edges, which contrasts with the prominent role edge detection is thought to play in vertebrate vision.

To understand how fish integrate visual information from both eyes, we tested fish responses when stimuli were presented simultaneously to each of the eyes (Fig. 2f, Fig. S4e and Movie 7). Here again, 7 dpf larvae tended to turn away from the side presented with the larger stimulus, yet the response tendency was attenuated compared to the case where the same stimulus was presented alone. We found that the response to two competing stimuli can be very accurately predicted by linearly adding the two competing responses (each driving the fish in a different direction) recorded for each stimulus alone (Fig. 2f, S4e). We also note that responses to sets of stimuli that have equal angular difference between the eyes (e.g. 36º vs 27º and 27º vs 18º) were markedly different from each other, yet the response to each set could be accurately predicted by linearly adding the individually recorded responses (Fig. S4e). Importantly, the attenuation caused by two competing stimuli did not seem to result in an increase in probability of forward swims, which would indicate averaging of stimuli between the eyes. In fact, when two equally large stimuli were presented to both eyes, the fish were equally likely to turn away from either the right or left stimuli, which is in line with a winner-take-all strategy for binocular integration rather than averaging[39] (Fig. S4f). These results indicate that the binocular integration of the stimuli is less likely to be computed at visual sensory areas, but rather at downstream areas responsible for the behavioral execution (see Fig. 5).

When we presented multiple stimuli together to both eyes we found, as expected, that averaging of responses to stimuli within an eye and the summation of the averaged responses between the eyes gave a very accurate prediction of the turning behavior of the fish (Fig. S4g).

Taken together, these results show that fish use different retina-wide computations to analyze visual occupancy in the different dimensions of the retina: they integrate visual occupancy in the vertical dimension, yet average over the horizontal one. Fish integrate visual information from both eyes using a winner-take-all strategy by probabilistically responding to retinal occupancy values from one of the eyes in each response bout.

**Older larvae use similar algorithms to respond to visual occupancy**. We next used the VR assay to explore the way 14 and 21 dpf larvae integrate and respond to visual occupancy, as fish at these ages begin turning towards their neighbors as opposed to the purely repulsive interactions at 7 dpf (Fig. 1d, f). For both these older age groups, we observed the emergence of attraction to projected stimuli of small angular size, in combination with repulsion from larger stimuli (Fig. 3a, b, "Methods"). At 14 dpf, the transition to repulsion occurs already for very small angular

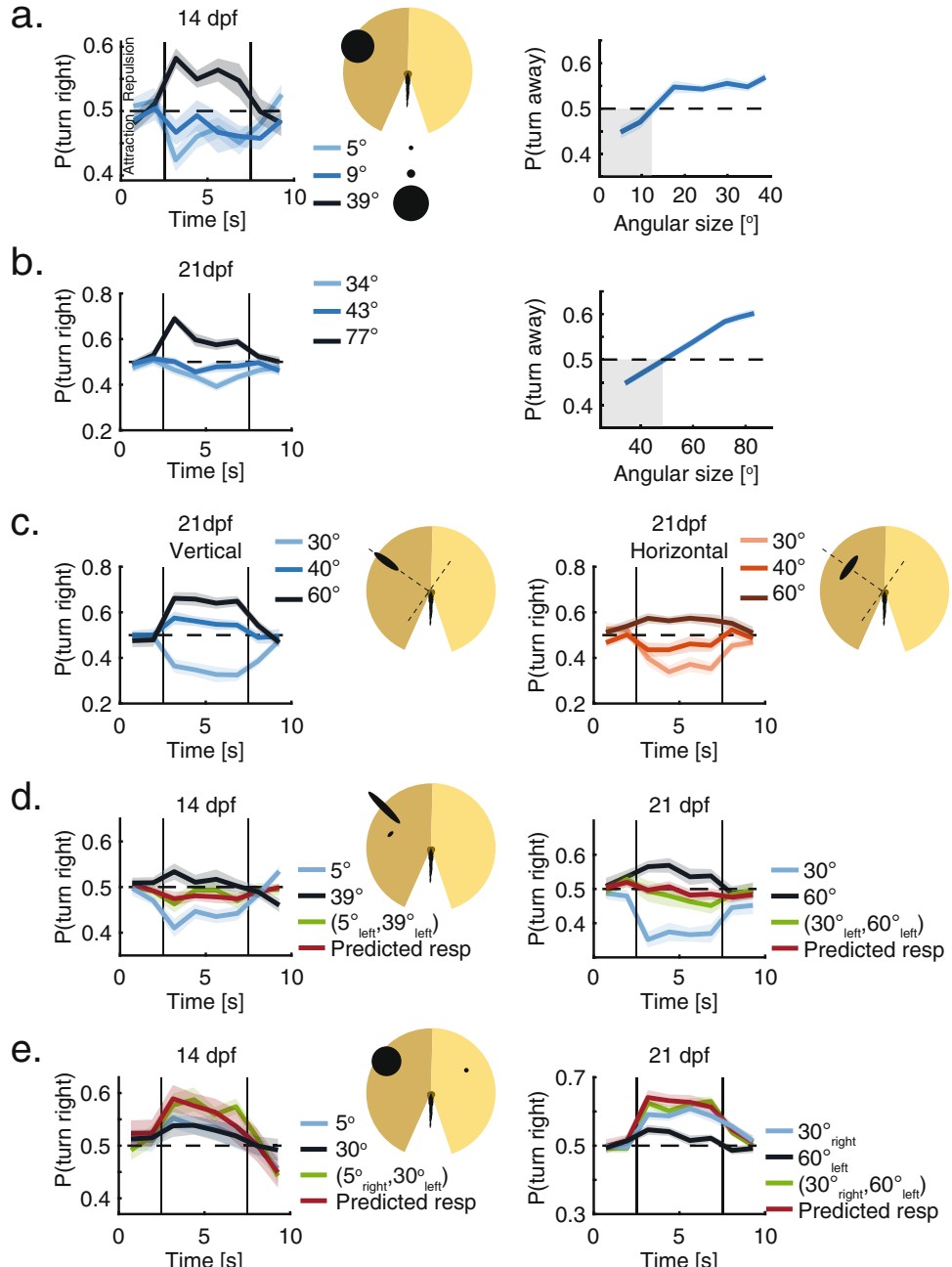

**Fig. 3 Older larvae use similar algorithms to integrate visual social information. a** Left: Probability to turn right per bout in response to dots of different sizes presented to the left visual field. At 14 dpf, fish show both attraction to small angular sizes and repulsion from larger sizes ($N = 32$ fish). Right: Probability to turn away from dots of different sizes, presented to the left of the fish, calculated over the entire stimulus duration. Gray shaded area marks retinal occupancy values for which fish exhibit attraction. **b** Same as in A, only for 21 dpf larvae ("Methods"). **c** Left: Probability of 21 dpf larvae to turn right per bout, in response to ellipses of increasing vertical size (perpendicular to the plane of the eye), while horizontal sizes remain constant at 23º ($N = 32$ fish). Right: Same but in response to ellipses of increasing horizontal sizes (parallel to the plane of the eye), while vertical sizes remain constant at 23º ($N = 32$ fish). **d** Left: Probability to turn right per bout in response to two images presented together to the left visual field of 14 dpf larvae (green line); to each of the images presented alone (blue lines) and the prediction based on the weighted average of the responses to each stimulus presented alone (red line, "Methods"), where the weights of the responses are equal ($N = 32$ fish). Right: Same but for 21 dpf larvae. **e** Left: Probability to turn right per bout in response to two dots presented simultaneously to both eyes of 14 dpf larvae (green line); for each dot presented alone (blue lines) and the prediction based on the linear summation of the recorded response biases to each dot presented alone (red line, "Methods")($N = 32$ fish). Right: same but for 21 dpf larvae. In all panels probability to turn right per bout is calculated as the fraction of right turns out of all turns in 1.25 s time bins ("Methods"). Bold lines are average; shaded areas are SEM.

sizes (>11º), while at 21 dpf, animals remained attracted to stimuli as large as 45º, and only turned away from even larger stimuli. This is in accordance with the ontogeny of aggregation behavior in group swimming experiments, in which 14 and 21

dpf larvae were increasingly attracted to their neighbors (Fig. 1d, f). Turning bias in older larvae, similar to observations in young larvae, was mostly modulated by the probability to turn in a certain direction, while the magnitude of the turns, the bout rate

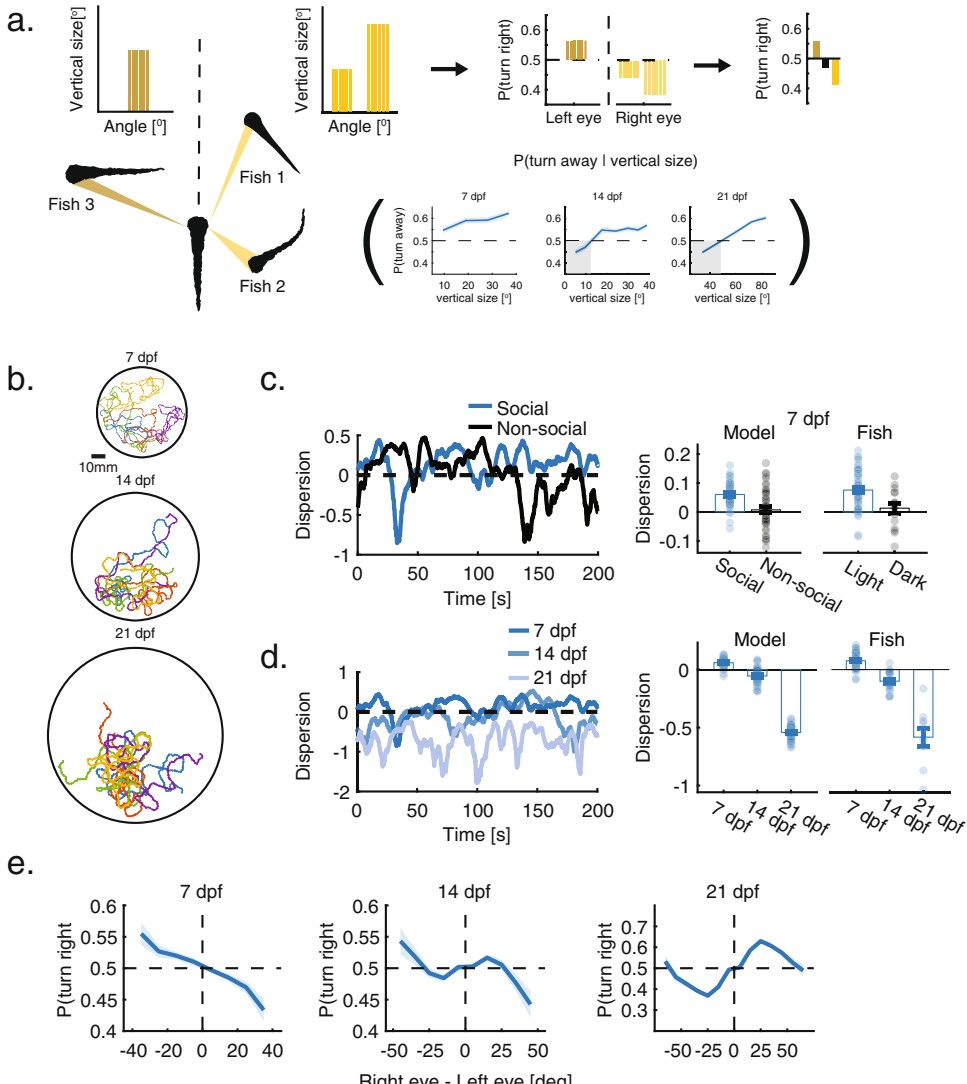

**Fig. 4 Social interactions extracted from VR capture the behavior of real groups. a** Models are based on the vertical visual occupancy casted by neighbors on the retina of the focal fish and the algorithms and response functions extracted from VR experiments (Eq. 1 and "Methods"). Left: Each neighbor is represented by its estimated vertical size at specific visual angels. Middle: The vertical sizes casted by neighbors elicit a turning bias based on the age dependent response functions observed in VR experiments $p(turn\ right|v_i)$. Right: The turning biases are then averaged on each side (yellow bars) and compared between the eyes to elicit a turing probability $p(turn\ right)$ (black bar) for the next bout performed by the fish ("Methods"). **b** Examples of simulated trajectories for 7, 14, and 21 dpf larvae in arena sizes that match real group experiments. **c** Left: Normalized dispersion values for one simulated group of 5 fish at 7 dpf, with and without social interactions. Zero represents the expected average dispersion when fish do not interact, and positive values represent overdispersed distributions ("Methods"). Right: Simulated groups of 5 fish at 7 dpf modeled with social interactions show higher dispersion levels than simulated groups modeled without social interactions. Bars represent means; error bars are SEM ($N = 50$ simulations). Experimental data of fish swimming in the light and in the dark is plotted for comparison (same as Fig. 1c). **d** Same as in (**c**), only for simulations of different age groups. Simulated groups switch from overdispersed to clustered groups from 7 to 21 dpf. Experimental data of different age groups is plotted for comparison (same as Fig. 1**d**). **e** Probability to turn right as a function of the difference in total angular occupancy experienced by each eye (negative values - higher occupancy to the left) in the simulations. (Similar to Fig. 1e, f for group experiments). Bold lines represent turning probability calculated as the fraction of right turns out of all turns collected from all simulated fish in 5º bins; error bars are the 95% confidence interval of the fitted Binomial distribution to the events in each bin.

and the average path traveled in a bout were only mildly affected by the size of the stimuli (Fig. S5a-b).

In line with observations at 7 dpf, the orientation of the stimulus on the retina had a marked effect on fish responses also at 14 and 21 dpf. An increase to the object's vertical dimension (i.e. height of the image on the retina) was largely responsible for the size dependent transition from attraction to repulsion in both age groups (Fig. 3c, S5c). In addition, we found that unlike the 7 dpf fish, these older animals were not agnostic to changes in the horizontal dimension (i.e. width of the image on the retina): here,

an increase to the width of the stimulus also contributed to its repulsive power, but to a lesser extent than an increase to its vertical dimension (Fig. 3c, S5c).

Integration of information from multiple stimuli presented together to one eye of the fish at 14 and 21 dpf, followed a similar algorithm to the one observed at 7 dpf. The responses to such joint presentation of stimuli could be accurately described by the weighted average of the recorded responses to each of the stimuli presented alone, even if the two stimuli elicited contradicting responses (attraction vs. repulsion) (Fig. 3d). Yet unlike the size

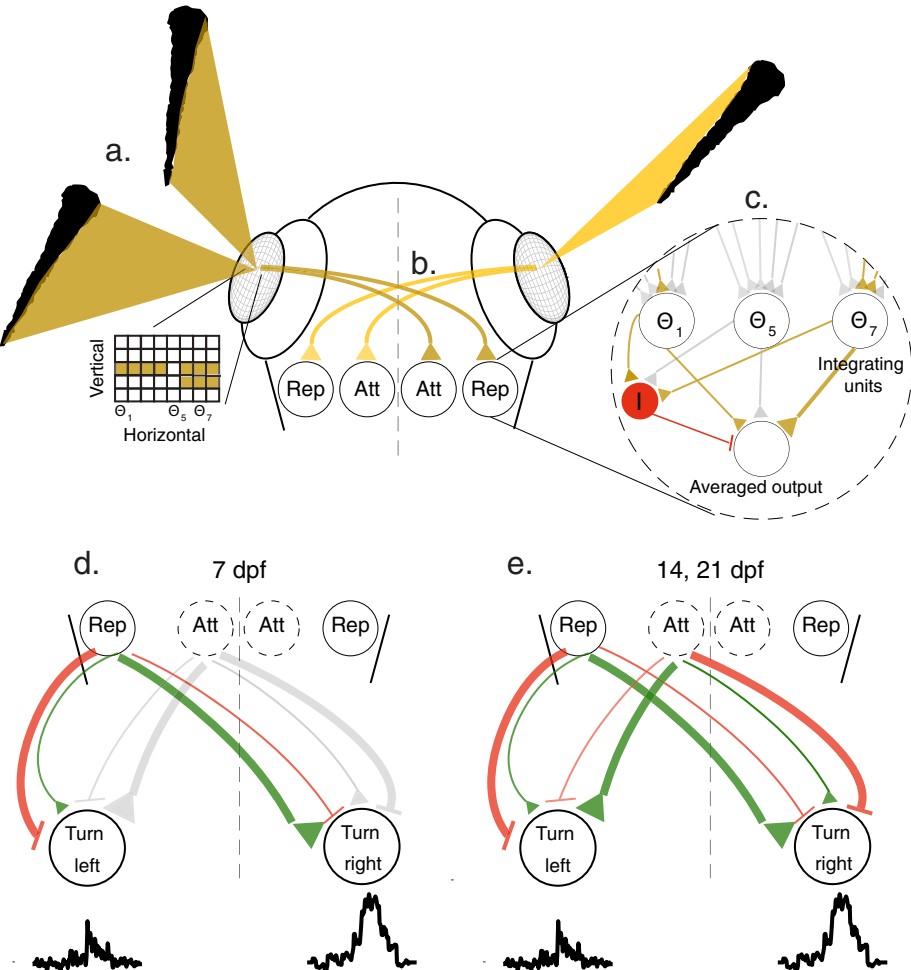

**Fig. 5 Conceptual circuit model describing visuomotor transformation underlying social interactions. a** Images casted by neighboring fish on the retina of the focal fish are represented as a two-dimensional grid of retinal ganglion cells mapping the vertical height of the neighbors at different viewing angles $\Theta$. **b** Retinal ganglion cells selectively project to two separate populations in visual areas: the 'Rep' population, which ultimately elicits turning responses away from the stimulated eye, and the 'Att' population, which elicits turning towards the stimulated eye at older ages. **c** The activity of all ganglion cells at a visual angle $\Theta_i$ are combined by specific integrating units to represent the vertical height of the stimulus at that angle. These units relay their activity to an output population (brown triangles show activation, gray triangles represent inactive connections); we show three such example units, which represent the corresponding visual angles in (**a**). Active integrating units also stimulate an additional inhibitory population I (red), which suppresses the output population, and performs averaging of the inputs from the integrator units (see text). **d, e.** Each population in the visual areas can excite (green) and inhibit (red) motor centers on the ipsi and contralateral sides. At 7 dpf (**d**) responses are dominated by repulsion from the stimulated eye, while at 14 and 21 dpf (**e**) the balance of both the attractive and repulsive populations will determine the turning direction of the fish. Line thickness represents the strength of the excitation/inhibition, gray color represents inactive connections.

dependent weighing of the stimuli at 7 dpf, equal weights to both stimuli gave the best prediction of the observed responses at 14 and 21 dpf. Such equal weighing indicates that larger stimuli that elicit repulsion do not take precedence over smaller stimuli eliciting attraction, and suggest that they might involve different visuomotor pathways (see Fig. 5).

When presented simultaneously with stimuli to both eyes, the algorithms for binocular integration observed at 14 and 21 dpf again followed closely those seen in younger larvae. The observed response to two competing stimuli was accurately predicted by the linear summation of the responses to each stimulus presented alone (Fig. 3e). Interestingly, this was true also in the case where one of the stimuli evoked repulsion and the other evoked attraction, resulting in an additive effect and a higher probability to turn in a certain direction than that of each stimulus on its own.

These results suggest that while social responses become more complex as larvae mature, and involve both repulsion from higher occupancy values and attraction to lower ones, the algorithms

used by 7 dpf larvae to integrate visual occupancy over the retina are largely conserved over development.

**Modeling collective swimming behavior based on responses to retinal occupancy.** We next tested whether social interactions based on the visual integration algorithms extracted from VR can accurately account for group behavior in larval zebrafish. To that end, we simulated groups of 5 or 10 agents (similar to the group swimming experiments described in Fig. 1, S1) that interacted according to these rules (Fig. 4a). In total, we simulated 4 variants of the model - a non-social model, in which fish do not interact with one another and 3 social models (one for each age) based on the visual integration algorithms and behavioral responses observed in VR at 7, 14 and 21 dpf (Fig. 4a, Movies 8–10). The simulated trajectories of the fish in all models were composed of discrete bouts and changes in heading direction, which were based on the swimming statistics extracted from group

experiments (Fig. S1d, S6a–c). In the social models, the fish biased their turning direction in each bout based on the visual occupancy that the neighboring fish cast on both eyes. Specifically, each occupied horizontal visual sub-angle $\theta_i$ on the retina, elicits a turning bias based on its integrated vertical size $v_i$: $bias(v_i) = p(turn\ right|v_i) - 0.5$ (positive values represent a rightward bias, negative is leftward), where the age relevant turn probabilities $p(turn\ right|v_i)$ were learned from VR experiments (Figs. 2c, 3a, b, right). Next, in accordance with the monocular and binocular integration algorithms observed in VR, these turning biases are averaged over all occupied visual angles $\theta$ on each side of the fish, and finally the (signed) average responses are linearly added, such that

$$p(turn\ right|\theta^{left}, \theta^{right}) = 0.5 + \sum_i^{\theta^{left}} w_i^{left} \cdot bias(v_i^{left}) \\ + \sum_i^{\theta^{right}} w_i^{right} \cdot bias(v_i^{right}) \quad (1)$$

where $w_i$ is the relative weight assigned to each response bias and represents either a weighted average $w_i = v_i/\Sigma v_i$ (7 dpf) or a simple average $w_i = 1/N_\theta$ (14 and 21 dpf) of the monocular turning biases (Fig. 4A). The intercept 0.5, centers the summed responses around that value and $p(turn\ right|\theta^{left}, \theta^{right})$ is bounded between 0 and 1 using a piecewise linear function (see "Methods"). Turning direction is then set probabilistically according to $p(turn\ right|\theta^{left}, \theta^{right})$ in that bout (with $p(turn\ left|\theta^{left}, \theta^{right}) = 1 - p(turn\ right|\theta^{left}, \theta^{right})$). Thus, all models use the same visual integration algorithms (Eq. 1) yet differ in the nature of the turning bias elicited by vertical visual occupancy $p(turn\ right|v_i)$ and the relative weights $w_i$ assigned to the responses (Fig. 4a). Importantly, the models have no free parameters that are tuned to the data. Each variant of the model was simulated 50 times to account for the inherent stochasticity of the models and the results of these simulations were trajectories of moving agents in confined arenas, similar to those extracted from real groups of fish (Fig. 4b, see "Methods" for a detailed description of the models). Finally, we tested the added benefit of characterizing social interactions using VR by comparing these models to an alternative set of social models (one for each age group), which were based on the visual interactions observed in group swimming experiments (Fig. 1f). This alternative set of models was similar in all individual fish swimming properties, yet the fish in these models modulated their turning directions according to the difference in total retinal occupancy between the eyes, which we estimated for all ages in the group swimming experiments (Fig. 1e, f, "Methods").

**Responses to retina-wide visual occupancy accurately predict the behavior of real groups of fish.** Simulated groups, based on the visual integration algorithms observed in VR at 7 dpf, showed an increase in group dispersion compared to the non-social model, which exhibited dispersion values that were at chance levels (Fig. 4c). These results capture well the behavior of groups of 7 dpf larvae swimming in the light and in the dark (Fig. 4c). Simulations of 14 and 21 dpf larvae generated an age dependent decrease in dispersion (or increase in group cohesion) quantitatively similar to the pattern observed in group swimming experiments, indicating that these interactions are sufficient to explain age dependent changes in group structure (Fig. 4d). The accuracy of the models in capturing group structure also generalized well to larger groups of 10 fish swimming together (Fig. S6d). Importantly, models that were based on the algorithms extracted from the VR assay (Fig. 4a) were more accurate in predicting average aggregation of groups than models based on

the visual interactions extracted directly from group swimming experiments (Fig. 1f). Average prediction errors in models based on group swimming experiments were 2.2, 1.06 and 126 times larger than those obtained by models based on VR for 7, 14, and 21 dpf fish respectively, indicating that the magnitude of the attractive responses at 21 dpf were severely underestimated in group swimming experiments (Fig. S6e). We note that simulated groups based on these minimal models did not exhibit an increase in group alignment as observed in real groups at 14 and 21 dpf, suggesting that alignment might involve additional processes not included in our models (Figs. S6f, S1b).

Our findings indicate that extracting social interactions directly from animal trajectories (as opposed to using VR) might hinder the identification of the correct interactions used by the fish (Fig. S6e). To further corroborate this finding, we attempted to extract the interaction rules used to create the simulations (Fig. 4a, Eq. 1) directly from the simulated trajectories. Specifically, we repeated the calculation that was used for real fish swimming in a group (Fig. 1e, f, "Methods") and estimated how the difference in total angular occupancy that simulated neighbors cast on each eye of the focal fish modulated its turning direction. We found that for simulations of 14 and 21 dpf fish, we could not retrieve the correct response functions used for the simulations, and that for 7 dpf, the inferred responses underestimated the strengths of repulsion (Compare Fig. 4e to Fig. 4a). Yet interestingly, these (inaccurate) response functions that we estimated from the simulated trajectories (Fig. 4e) very closely resembled the response functions extracted from group swimming experiments (Fig. 1f), which themselves did not give an accurate description of the emergent group behavior. This further emphasizes that using VR can provide a more accurate description of the actual algorithms used by interacting fish.

**A conceptual model of the underlying neurobiological circuits.** The specificity of the behavioral algorithms extracted from the VR experiments allows us to make explicit predictions about the underlying neural circuits in the visuomotor processing stream. We therefore propose a conceptual circuit model, depicted in Fig. 5, which transforms visual occupancy on the retina into behavioral decisions.

This model takes the visual occupancy elicited by neighbors on each retina as the sole input and represents it as a two-dimensional ensemble of activated 'retinal ganglion cells', operating as dark detectors in this case, as neighboring fish are expected to be mostly darker than the background (Fig. 5a, inset). These visual inputs are relayed to downstream visual areas (e.g. optic tectum)(Fig. 5b), where a retina-wide integration of the vertical dimension is performed, thereby compressing the two-dimensional grid of the retina into a one-dimensional array of neurons representing the integrated values at each horizontal viewing angle ('Repulsive' population) (Fig. 5c). Next, the activity across this one-dimensional array of cells is averaged to generate a single output value for each eye, which represents the size selective tendency of the fish to turn away from the visual occupancy presented on the 2D retinal grid. Such averaging can be achieved by an additional inhibitory input to the integrating units, where the suppression is inversely proportional to the number of visual angles activated on the retina (akin to divisive normalization[60–62] (Fig. 5c).

At later stages in development (14 and 21 dpf), we propose that a second circuit module emerges, which responds maximally to low vertical occupancy and reduces its activity as vertical occupancy grows ('Attractive' population)[63–65] (Fig. 5b). This module, by similar means, also generates a single output value for each eye, which induces an inverse size selective attraction

towards low vertical occupancy. The output values from both circuit modules then excite/inhibit units in downstream areas, probably in the hindbrain, where lateralized activity is known to be responsible for controlling directed turns of the fish[43,44] (Fig. 5d, e). At 7 dpf, these visuomotor connections are dominated by contralateral excitation and/or ipsilateral inhibition from the visual occupancy integrating neurons ('Repulsive' population) to elicit competition between the two lateralized hindbrain regions, and finally a turning response away from the more occupied eye (Fig. 5d). At 14 and 21 dpf, the additional population activated by low vertical occupancy on the retina ('Attractive' population) elicits an opposite response, by ipsilateral excitation and/or contralateral inhibition, which results in turning towards the stimulated eye (Fig. 5e). At these older ages, the attractive and repulsive tendencies from both eyes will then compete (or add up) to elicit the observed attractive and repulsive responses of the fish.

The specific elements in this hypothesized model, e.g. units that represent integrated vertical occupancy and averaged horizontal occupancy in visual areas, excitation/inhibition of units in the contra/ipsilateral side in the hindbrain and even the emergence of additional modules over development can be readily tested, rejected or refined using whole-brain imaging and connectivity data from real fish[66,67].

## Discussion

Here, we combined observations of freely swimming groups of fish with targeted manipulations of visual inputs using a VR assay, and simulations of minimal models of collective behavior to identify the specific algorithms that govern visually based social interaction from ages 7 to 21 dpf. Our results show that larval zebrafish exhibit collective group structure already at 7 dpf and perform complex computations based on integrated retinal occupancy as the input to the animal. Importantly, the basic algorithms that allow fish to integrate and respond to social visual inputs at 7 dpf were largely conserved over development, even though the repertoire of the responses to neighbors was expanded to include both attraction and repulsion at 14 and 21 dpf, as opposed to only repulsive interactions at 7 dpf. Using model simulations, we were able to show that the behavioral algorithms observed in VR experiments can very accurately describe group structure over development, which highlights the necessity of using such assays. Our findings allowed us to hypothesize the structure of the neural circuits underlying these behavioral algorithms. These predictions can be readily constrained, rejected or validated in future experiments, which combine our established virtual social assay with neural recordings.

Our results indicate that fish integrate visual occupancy in the vertical dimension of the retina, use spatial averaging in the horizontal dimension and inter-eye competition based on a winner-take-all strategy to decide on the direction of their next movement. Behavioral algorithms that combine stimulus averaging and winner-take-all strategies, together with their neural substrates, were previously reported in larval zebrafish when escaping from threatening stimuli[39]. The observed responses to social stimuli reported here are quantitatively and qualitatively different from escape behaviors, therefore it will be interesting to explore the similarities and differences between the brain areas and neural circuits involved in social interactions compared to those reported for escape behaviors.

Previous experimental studies and models of collective behavior implied that animals execute complex operations such as object classification, distance measurements or object counting and that the results of these operations are available to the animals for further processing. Here we found that larval zebrafish use a much simpler strategy, namely retina-wide integration of visual occupancy, which does not rely on any such complex operations. These findings are in line with recent theoretical models suggesting that raw visual inputs are sufficient to elicit complex collective behaviors[10,27,28]. Notably, behavioral algorithms based on retina-wide integration of visual inputs are expected to fail when fish perform other behaviors such as hunting for example, which specifically requires object classification prior to any further behavioral executions. We expect such behaviors to rely on different neural circuits than the ones used for social interactions.

Interestingly, we found that visual occupancy in the vertical dimension across the retina was the dominant input affecting behavior at 7 dpf, while occupancy in the horizontal dimension was largely ignored. Due to a neighboring fish's elongated shape, the horizontal extension of its projection on the retina will depend strongly on its orientation with respect to the observer. The vertical projected size, on the other hand, is less variable as it is independent of the neighbor's orientation and only depends on its distance. We hypothesize that this is the reason why young larvae integrate only over the vertical dimension to guide their turning responses.

A monotonic response to vertical size predicts a maximal effect of objects spanning the entire retina. Such a behavioral algorithm will fail when fish encounter many forms of aquatic vegetation for example. Since larval zebrafish do not avoid such areas, but frequently seek them for protection, we propose that the avoidance mechanism is constrained to objects with a limited vertical size. Such spatial filtering of visual statistics is well described in hypercomplex and end-stopped neurons in the mammalian visual cortex[68–70], and their presence in the zebrafish visual system awaits confirmation by further studies.

At 14 and 21 dpf, the vertical dimension of the retinal image was still the dominant dimension eliciting repulsion, yet fish also responded to the horizontal dimension of the image comprising more complex responses. As this increase in complexity develops together with an increase in group alignment, we hypothesize that it might represent the developing tendency to detect and respond to the body orientation of neighbors as an additional input to the fish. Future experiments using VR assays as we used here, can specifically test if older larvae or juvenile fish are capable of detecting and responding to neighbors' body orientation and motion or if they are largely agnostic to it[22,23,25,26,71].

Attraction to low visual occupancy was observed only in older larvae. We hypothesize that the neural circuits that support attraction behavior develop with age. However, the lack of attraction to smaller angular sizes at 7 dpf, can also stem from a limitation of the developing visual system, where retinal ganglion cell receptive field size decreases with age[72]. Still, the increased tendency to repel from objects of increasing angular sizes at 7 dpf and the tendency of older larvae to attract to these same angular sizes, supports the notion of a developmental 'switch' in the tendency to attract to neighbors, that cannot simply be explained by a developmental change in size tuning of retinal receptive fields. Interrogation of the neuronal responses to virtual neighbors in future studies can specifically characterize the capabilities of young animals to resolve small object sizes, and to detect nascent circuits responsible for attraction if and when they develop.

The social responses observed in group swimming experiments and the responses we probed using the VR assay were based solely on visual input. Previous studies showed that larval zebrafish also use non-visual cues, such as mechanosensory[33,56] and chemical stimuli[32] for social interactions. In this study, we did not test how different sensory modalities operate jointly to support collective behavior. It will be interesting to test how visual information at

longer distances is supported by mechanosensory sensation at shorter distances to elicit social responses[33], or how visual social information is related to chemical stimulation that represents conspecifics[32]. These combinations can now be tested in future studies.

Our findings represent an important step toward elucidating the neural circuits and mechanisms at the basis of collective social behavior. First, we have detected robust computations already present at 7 dpf, a critical age in which the entire nervous system of the fish is easily accessible via functional imaging techniques at single cell resolution[48–50,52]. In addition, we find that the basic algorithmic components we uncovered are mostly conserved during development, indicating the possibility that the underlying neural circuits are relatively matured already at 7 dpf. Second, using VR we identified the relevant dimensions of the visual stimuli that affects behavior and the underlying algorithms that transform visual stimuli into the observed movement responses. The specificity of these algorithms allowed us to hypothesize the circuit elements involved in these computations and to make testable predictions about their structure. Performing whole-brain imaging in a similar experimental assay will allow us to test, reject and refine these hypothesized circuit models, and to gain novel insight into the neural mechanisms underlying collective social behavior.

## Methods

**Fish husbandry**. All larvae used in the experiments were obtained by crossing adult AB zebrafish. Larvae were raised in low densities of approximately 40–50 fish in large petri dishes (D = 12 cm). Dishes were filled with filtered fish facility water and were kept at 28 °C, on a 14–10 h light dark cycle. From age 5 dpf, fish were fed paramecia once a day. On day 7, fish that were not tested in behavioral experiments, were returned to the fish facility where they were raised in 2 L tanks filled with 1.5″ nursery water (2.5 ppt), with ~15 fish in each tank and no water flow. On days 10–12 water flow was turned on and fish were fed artemia 3 times a day until they were tested at 14 or 21 dpf. All experiments followed institution IACUC protocols as determined by the Harvard University Faculty of Arts and Sciences standing committee on the use of animals in research and teaching.

**Free-swimming experiments**. Fish were transferred from their holding tanks to custom-designed experimental arenas of sizes d = 6.5, 9.2, 12.6 cm, depending on the age of the fish (7, 14, and 21 dpf respectively) filled with filtered fish facility water up to a height of ~0.8 cm. Experimental arenas were made from 1/16″ clear PETG plastic and had a flat bottom and curved walls (half a sphere of radius 0.5 cm) to encourage fish to swim away from the walls. Arenas were sandblasted to prevent reflections. Every experimental arena was filmed using an overhead camera (Grasshopper3-NIR, FLIR System, Zoom 7000, 18–108 mm lens, Navitar) and a long pass filter (R72, Hoya). All experimental arenas were lit from below using 6 infrared LED panels (940 nm panels, Cop Security) and from above by indirect light coming from 4 32 W fluorescent lights. Every 4 cameras were connected to a single recording computer that recorded 4 MP images at 39 fps per camera. To prevent overload of the RAM we performed online segmentation of the recorded images and saved only a binary image from the camera stream. The segmented images were then analyzed offline to extract continuous tracks of the fish using the tracking algorithm described in ref. [29]. All acquisition and online segmentation were performed using costume designed software written in Matlab. Every group was imaged for ~5 minutes, after fish were allowed 5–10 min to acclimate to the arena. Groups were eliminated from subsequent analysis in the case that one or more of the fish were immobile for more than 25% of the experiment. All and all 35%, 22%, and 33% of groups ages 7, 14, and 21 dpf were eliminated from the analysis due to immobility of the fish. Choosing a more stringent, or a less stringent criteria for elimination did not change the qualitative nature of the results.

**Individual and group properties of free-swimming fish**. The position of each fish $i$ at time $t$ is defined as the center of mass of the fish extracted from offline tracking and is denoted as $\overrightarrow{x_i}(t)$. The velocity of each fish $i$ is given by $\overrightarrow{v_i}(t) = [\overrightarrow{x_i}(t + dt) - \overrightarrow{x_i}(t - dt)]/2dt$, where $dt$ is 1 frame or 0.025 s. The speed of the fish is then $S_i(t) = |\overrightarrow{v_i}(t)|$, and the direction of motion is $\overrightarrow{d_i}(t) = \overrightarrow{v_i}(t)/|\overrightarrow{v_i}(t)|$.

For the group, we calculated a normalized measure of group dispersion: $Dispersion(t) = log(NN_1(t)/NN_1^{shuffled})$ where $NN_1(t)$ is the average nearest neighbor distance and $NN_1^{shuffled}$ was calculated from randomized groups created by shuffling fish identities such that all fish in a given randomized group were

chosen from different real groups. Positive dispersion values mean that real groups are more dispersed than shuffled controls and 0 means equality. Group alignment was defined as $alignment(t) = |\sum_i^N \overrightarrow{d_i}(t)|/N$, where N is the number of fish in the group. Chance levels were similarly calculated from randomized groups (see above), and alignment values are bounded between 0 (all fish are facing in different directions) and 1 (fish are completely aligned).

**Estimating retinal occupancy using ray casting**. To estimate the visual angle that each neighbor in the group casts on the eye of focal fish $i$, we used a modified ray casting algorithm[57,58]. Specifically, we casted 1000 rays from each eye of the focal fish spanning 165º from the direction of motion towards the back of the fish, leaving a total of 30º of blind angle behind the fish. This amounts to an angular resolution of ~0.165º per line. We then detected all pixel values representing fish in the paths of the rays and calculated the visual angle occupied by each fish and the total occupied visual angle experienced by each eye (Fig. 1e).

**Segmenting fish trajectories**. Trajectory segmentation into discrete bouts or decision events was done by detecting local minima points in the speed profile of the fish[29]. A bout was defined as the motion between two consecutive local minima. Individual events were then characterized by the duration of the event, the total path traveled and the change in heading direction, or turning angle, between the start and the end of the event.

**Turning in response to the arena walls**. To estimate how the walls of the arena affect the turning behavior of the fish we calculated the probability of the fish to turn in a certain direction for a given distance ($D_{wall}$) and direction (left/right) of the closest wall: $P(turn\ right|D_{wall})^{left/right}$ (Fig. S6c). Distance to the wall was grouped into bins of 1 body length, and turning probability was calculated as the fraction of right turns out of all turns, recorded from all fish, in each bin. Error bars represent the 95% confidence interval of the fitted Binomial distributions to the data in each bin. Responses to the wall seem to decay to chance levels at distances > 3 body length.

**Turning in response to the difference in visual occupancy between the eyes**. We estimated how the difference in total visual occupancy between the eyes $\Delta visual\ occupancy$ (see above) affects the binary turning direction (either left or right) of fish swimming in a group (Fig. 1f). Specifically, we calculated $P(turn\ right|\Delta visual\ occupancy)$ as the fraction of right turns out of all left/right turns recorded for 5º bins of $\Delta visual\ occupancy$ and estimated the 95% confidence interval from the fitted Binomial distribution to the data in each bin. We discarded all turning events at distance < 3 body length from the wall, as not to confound wall avoidance with neighbor responses. Data are pooled over all fish.

**Effects of total retinal occupancy on bout rate in groups of fish**. We estimated how the total visual occupancy in both eyes $\Sigma visual\ occupancy$ affects the probability of the fish to perform a bout or a movement decision (Fig. S1e). Specifically, we estimated $P(bout|\Sigma visual\ occupancy)$, which is the fraction of recorded bouts out of all events (bouts and idle times) in 5º bins of $\Sigma visual\ occupancy$ and estimated the 95% confidence interval from the fitted Binomial distribution to the data in each bin. Dividing by $\Delta t$, which is our sampling resolution allows us to transform bout probability into bout rates. We discarded all bouts at distance <3 body length from the wall, as not to confound wall avoidance with neighbor responses. Data are pooled over all fish.

**Virtual reality assay**. We combined the experimental system that was used to track groups of fish (see above) together with bottom projection of visual stimuli in closed loop as our virtual reality assay (Fig. 2a). In all experiments, a single fish in each arena interacted with images projected directly onto the sandblasted flat bottom of the arena (diameter = 9.2 cm)(Fig. S3A). All fish tracking and posture analysis were done using custom software written in Python 3.7 and OpenCV 4.1 as described in ref. [43]. Briefly, movie images acquired at 90 Hz were background subtracted online to obtain an image of the swimming fish, and body orientation was estimated using second-order image moments. We used the specific position and body orientation of the fish to present moving images that are locked to the position and heading direction of the fish (Movies 6–7). We defined swim bouts using a rolling variance of fish orientation (50 ms sliding window) with a constant threshold. Visual stimuli were presented only when fish were stationary and were turned off during swim bouts.

**Visual stimuli**. Images were presented on one or both sides of the fish. Stationary images appeared at a constant angular position (±50° from the heading of the fish) and radial distance (0.825 cm to the closest edge of the presented image) with respect to the fish, and stayed on while the fish was stationary and until the end of the trial. Different trials were separated by an inter-stimulus interval (ISI) equal in length to the time of stimulus presentation (5–5.6 s). The temporal order of different stimuli was randomly shuffled during an experimental session.

Moving images appeared at a constant radial distance (0.825 cm to the closest edge of the image) in the periphery of the visual field (±60 with respect to the fish's direction of motion 0°) and moved towards the center of the visual field ( ±30°) in bouts mimicking fish natural motion (Movies 6–7).

For 21 dpf larvae, due to the larger range of angular sizes needed to probe the behavior of the fish, visual occupancy was modulated by positioning a constant size image (0.8 cm in diameter) at different radial distances (instead of changing the size of the image presented at a constant distance, as was done for 7 and 14 dpf fish). When more than one image was presented to the same visual field of the fish, unless otherwise stated, the images were separated by empty space equal to the width of the presented images.

In all experiments, every stimulus or stimulus combination presented to the fish, always had a mirror image stimulus presented to the fish on a separate trial. In all analyses presented throughout the manuscript, trials with mirroring stimuli are flipped and combined together.

**Virtual interactions in a bowl-shaped arena.** We projected images (on one or both sides of the fish) onto a half dome shaped arena ($R = 3.6$ cm) made from commercially available light diffusers (*Profoto*). Domes were filled with water to the top, and projected images were centered at the mid-level of the dome, i.e. ~1.8 cm from the bottom. Projected images were corrected to account for the curvature of the dome to eliminate distortions. We used stationary stimuli situated at $\pm 60°$ from the fish's heading direction. We adjusted the sizes of the projected images depending on the distance of the fish from the walls, such that the estimated angular sizes of the images on the retina were constant for a given trial. The maximal angular size we used was 18° to avoid images becoming too large when the fish is far from the wall. We did not present images when the fish's distance from the center of the arena was larger than the distance of the middle of the projected image from the center of the area. In these cases, the fish was too close to the walls and we could not estimate the size of the projected image.

**Measuring virtual interactions between fish swimming in separate tanks.** We tracked the positions of four individual fish, each swimming in separate identical arenas (D = 9.2 cm). We then projected three moving dots (D = 0.3 cm) in each of the separate arenas, which exactly mimicked the position and velocity of the three fish swimming in the other arenas (Fig. S2a)[31]. Every experiment consisted of 60 trials, and each trial consisted of 60 s where the dots mimicking neighbors were visible to the focal fish ('on') and 30 s where the dots were not visible ('off'). We then combined the tracked positions of the four real fish from the separate arenas and analyzed them as a single 'virtual' group (Fig. S2).

**Retina model.** To estimate the shape, size and position of the projected object's image on the retina of the fish, we used a pinhole model of the retina (Fig. S7a). A toolbox in python implementing this model can be found at: https://github.com/nguyetming/retina_model.

Specifically, we used the following parameters when modeling the retina of the larval zebrafish (Fig. S7b): distance between the eyes: 1.2 mm, eye radius: 0.45 mm, average height of the fish above the projection plane (h): 5 mm, effective retina field: 163°[73], mean vergence angle: 36°.

**Quantifying fish responses to projected visual stimuli.** To analyze fish responses to the presented stimuli we calculated for each bout the change in body orientation of the fish, and the path traveled in that bout. We then calculated, for each fish, the time binned responses associated with these discrete bouts over all trials presenting the same stimulus ($N_{trials} = 40$ for each stimulus type). Specifically, we used all detected bouts in a given time bin and calculated the probability to turn right as the fraction of right turns out of all turns (e.g. Fig. 2c), the average and cumulative turning angles (e.g. Fig. S3h), the average path traveled in a bout and the bout rate of the fish (Fig. S3i).

**Predicting fish responses to combined visual stimuli presented to a single eye.** For 7 dpf larvae, we were able to accurately predict fish responses to two stimuli presented together to a single eye as the weighted average of the response biases elicited by each stimulus presented alone (Fig. 2e, S4c-d):

$$p_{predicted}(turn\ right | v_{left}^1, v_{left}^2) = bias(v_{left}^1) \cdot w^1 + bias(v_{left}^2) \cdot w^2 + 0.5$$

where $bias(v) = p(turn\ right | v) - 0.5$ (positive values are rightward biases and negative leftward), $v$ is the vertical dimension of the stimulus, $w^1, w^2$ are weights representing the relative sizes of the stimuli such that $w^i = v^i / \sum v^i$. The intercept 0.5, centers the averaged response around that value such that $p_{predicted} = 0.5$ means no bias in the turning direction. For 14, and 21 dpf $w^1 = w^2 = 0.5$ (a simple average) gave the best fit to the data (Fig. 3d). We speculate that the equal weighing at 14 and 21 dpf is due to the contradicting nature of the stimuli (attractive and repulsive responses) and might represent separate neural pathways (Fig. 5).

**Predicting fish responses to combined visual stimuli presented to both eyes.** We were able to accurately predict fish responses to two stimuli presented simultaneously to both eyes based on the recorded response biases elicited by each stimulus presented alone (Fig. 2f, 3e and Fig. S4e):

$$p_{predicted}(turn\ right | v_{left}, v_{right}) = bias(v_{left}) + bias(v_{right}) + 0.5$$

where $bias(v) = p(turn\ right | v) - 0.5$ is positive if the stimulus elicits a rightward bias and negative otherwise. The intercept 0.5 centers the summed biases around that value. For the cases of 14, and 21 dpf fish, in which the two biases can elicit a stronger turning response than that of each stimulus presented alone (Fig. 3e), the probability is bounded between [0 and 1] using a piecewise linear function that maps any values larger than 1 or smaller than 0 to these bounds. Such bounded response strength is akin to a biological ceiling effect, in which one stimulus is so strong that the addition of another doesn't add linearly to the animal's response. In practice, our fish did not reach such ceiling effects for any of the stimuli sizes and combinations we have tested.

**Modeling groups of free-swimming fish.** We simulated groups of N fish ages 7, 14, and 21 dpf, swimming in bounded arenas of diameters 6.5, 9.2, and 12.6 cm (respectively), interacting according to the algorithms observed in VR experiments (Fig. 4a, Movies 8-10) or according to the response functions estimated from free-swimming experiments (Fig. 1f).

a. *Bout size and rate.* In all simulations, each stationary fish, at every time step, probabilistically decides to perform a bout according to the average bout rate observed in group swimming experiments (Fig. S1d). Bout magnitude and bout duration followed that of the average bout calculated from real fish data (Fig. S6B).

b. *Wall interactions.* When simulated fish were at a distance <2BL from the walls, they turned away from the wall with probability drawn from the empirical responses of real 7 dpf fish swimming in a group (Fig. S6c). If the executed bout was expected to end outside of the arena, it was truncated to ensure the fish stays inside the simulated arena. When simulated fish were at a distance <2BL, they did not respond to their neighbors regardless of the model used.

c. *Non-social model.* Simulating N fish that perform wall avoidance at close distances as described above, and otherwise choose a new heading direction in each bout by randomly drawing a turning angle from the experimentally recorded turning distributions (Fig. S6a) constitutes the non-social model. These fish do not interact in any way.

d. *Social models based on the visual integration algorithms extracted from VR.* We used the algorithms we extracted from the VR assay to simulate the interactions between fish in the group according to Eq. 1 (see main text) and the recorded response functions to vertical visual occupancy extracted from VR experiments (Fig. 4a). In all models, we use the simulated height ($H_j$) and distance ($d_j$) of neighbor $j$ to calculate the vertical occupancy at visual sub-angle $i$ ($Vc_{ji}$) casted on the retina of the focal fish by that neighbor: $Vc_{ji} = 2 \cdot arctan(H_j/d_j)$. For simplicity, we did not account for occlusions among neighbors in estimating visual occupancy as initial simulations showed that it did not make a noticeable difference for the group sizes used here. In addition, we also assume that the height of the fish along its body axis is constant allowing us to treat the vertical occupancy at all visual sub-angles occupied by neighbor $j$ as a single value. The relative weight $w_i$ assigned to the the turning bias elicited by vertical occupancy at visual sub-angle $i$ casted by neighbor $j$ $bias(v_{ji})^{left/right}$(Eq. 1) within each eye followed the weights that best describes the responses of the fish to combined monocular stimuli in the VR experiments (Fig. 2e, Fig. 3d): a weighted average of the responses at 7 dpf, where weights are the relative vertical sizes $w_i = v_i / \sum v_i$ and a simple average of the responses at 14 and 21 dpf $w_i = 1/N_\theta$ where $N_\theta$ is the number of occupied visual angles ($\theta$) in a given eye. Since the response bias is equal along all visual sub-angles occupied by a given fish, we can simplify the implementation by averaging the responses across fish instead of across visual sub-angles.

e. *Social models based on visual integration algorithms extracted from group swimming experiments.* In these models we used the response functions extracted directly from group swimming experiments to simulate the social interactions of the fish. We calculated the visual angle of each neighbor on the retina of the fish using its width ($W_i$), distance ($d_i$), and relative orientation ($O_i$) to the focal fish. Specifically, we calculated the angle between the vectors pointing from the focal fish to the position of the head and tail of the simulated neighbor. We then summed all visual angles of neighbors within each eye and calculated the difference in occupancy or retinal clutter between the eyes. Here again we did not account for occlusions among fish, as initial simulations showed that it did not make a noticeable difference in the group sizes tested here. The summed visual angles within each eye were then used to calculate the probability to turn in a certain direction given the difference in visual occupancy between the eyes, $P(turn\ right | \Delta visual\ occupancy)$, using the inferred response functions from

group experiments (Fig. 1f). All other parts of the models are as described in **a**–**b** above.

f. *Model parameters used in simulations.*

| Parameter name | Description | Values (7,14,21 dpf) |
|---|---|---|
| Arena diameter | Similar to the arena sizes in group swimming experiments | 6.5, 9.2, 12.6 cm |
| Time interval ($\Delta t$) | Time between simulated steps | 1/50 s for all |
| Simulation time | Total simulated time per group | 600 s for all |
| Number of repetitions | Random repetitions of a given model | 50 for all |
| Fish starting positions | Random positions within 0.9*arena diameter | Same for all |
| Fish length | Estimated from fish | 0.4, 0.5, 0.8 cm |
| Fish height | Estimated from fish | 0.2, 0.25, 0.4 cm |
| Bout Rate | Estimated from group swimming experiments | 1.65, 1.4, 1.4 Hz |
| Bout size | Estimated from group swimming experiments | 0.1, 0.12, 0.16 cm |
| Bout duration | Estimated from group swimming experiments | 320 ms |

All modeling codes can be found at: https://github.com/harpazone/Modeling-larvae-social-behavior.

**Sample sizes, trial numbers and power estimation.** For all group swimming experiments we used sample sizes that were large enough to estimate group statistics (e.g. dispersion and alignment) according to previously reported data on collective behavior in zebrafish[20,29,53] and to also allow at least 25 degrees of freedom when parametric statistical models were used to compare between experimental conditions. We also chose to test two different group sizes (5 and 10 fish in a group) to test the generality of our findings. In the virtual reality assay, we used 40 trials per stimulus as this number proved sufficient to estimate the response of a single fish to the presented stimuli and 24–32 fish were used per experiment as our preliminary data showed that these are sufficient to estimate the mean responses of fish to the presented stimuli and the differences between these responses for different stimuli.

**Statistical testing.** We used parametric statistical models (one and two sample *t* tests, and one-way ANOVA) to compare experimental conditions, and mean and SD are reported in the text. Such parametric procedures assume normality of the hypothetical sampling distributions, which can be assumed due to the relatively large samples sizes in our analyses. Where applicable, we have tested for homogeneity of variances prior to conducting the statistical procedures and no deviations from the assumption of homogeneity were detected. As a measure of effect size, when comparing two independent groups we calculated Cohen's d statistic: *Cohen's d* $= \frac{\bar{x}_1 - \bar{x}_2}{Sp}$, where $\bar{x}_1, \bar{x}_2$ are the means of the two groups and $Sp$ is the pooled estimate for their standard deviations. All *p*-values reported are two sided, and no correction for multiple comparisons were employed in any analysis.

**Reporting summary.** Further information on research design is available in the Nature Research Reporting Summary linked to this article.

## Data availability

All raw data used in this manuscript can be found at: https://doi.org/10.7910/DVN/POVJYS.

## Code availability

A toolbox written in python implementing the pinhole retina model can be found at: https://github.com/nguyetming/retina_model. A software implementing larvae zebrafish social interactions can be found at: https://github.com/harpazone/Modeling-larvae-social-behavior.

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

## Acknowledgements
The authors thank all members of the Engert lab for support and advice during the project. We also thank Hanna Zwaka, Andrew Bolton, Mariela Petkova and Kristian Herrera for improving the manuscript and its visual content. Roy Harpaz received funding from Harvard Minds Brain and Behavior initiative. Florian Engert received funding from the National Institutes of Health (U19NS104653, R43OD024879, and 2R44OD024879), the National Science Foundation (IIS- 1912293), and the Simons Foundation (SCGB 542973). Armin Bahl acknowledges support from the Deutsche Forschungsgemeinschaft (German Research Foundation) under Germany's Emmy Noether Program (BA 5923/1-1) and Excellence Strategy (EXC 2117-422037984) as well as from the Zukunftskolleg Konstanz.

## Author contributions
R.H. and F.E. designed research, R.H., M.N.N., and A.B. performed research, R.H., M.N.N., A.B., and F.E. analyzed the data, R.H. and F.E. wrote the paper.

## Competing interests
The authors declare no competing interests.

## Additional information

**Peer review information** *Nature Communications* thanks Pawel Romanczuk, Tom Baden and the other anonymous reviewer(s) for their contribution to the peer review this work. Peer reviewer reports are available.

