## [Peer Review File · Nature Communications]

Reviewers' Comments:

Reviewer #1:

Remarks to the Author:

In this work the authors investigate what features of the visual input affect certain aspect of collective behavior (dispersion versus cohesion). The research is primarily experimental but contains also a modeling component. The research addresses a timely and novel questions with some state of the art methods like the usage of virtual reality essays in animal behavior: How does collective behavior in fish emerge from sensory inputs? How do individuals integrate visual inputs from their neighbors and respond to it with movement?

The paper is well written and comprehensible (for most part, see detailed comment below), the figures are of good quality. The results are novel and interesting, in fact also quite surprising like the role of the vertical extension of the visual projection. Overall, this result are of potential interest to scientists working on experiments and modeling of collective animal behavior and neuroscience, as well neighboring disciplines, and may thus may have some broader impact. Thus overall I think the paper may be potentially suited for Nat Comms. However, I see some issues that need to be addressed first:

1) If I understand the authors correctly, their result suggest that there is difference in response between moving or non-moving stimuli. This would indicate that optical flow does not seem to play a role for the turning response, which indeed appears surprising and should be discussed in more detail.

2) There seems to be inaccuracies in the terms used in the quantification of the continuous time stochastic process. In figures throughout the paper, the authors show curves for probability for turning right versus time. This is however ill defined. In continuous time stochastic processes a probability has to be defined with respect to a time interval, i.e. it is a product of a probability rate and a time interval. So either the authors implicitly binned time (e.g. according to the frame rate of recordings), which should be clearly stated, they mean in fact probability rates (which is unlikely given the numerical values) or something is odd here.

Of course one could consider individual bouts, and then for each bout (a discrete event) define a probability to turn right. In this case the probability rate would be then set by in the probability rate of single bouts, but this does not seem to be what is meant here.

3) the model description and parametrization in the main text and in particular in the methods is too short, lacking some important information, which make it hard to really understand what the model does, in particular for non experts:

- From just reading the section containing eq. 1 in the main text its not clear what the index i is? Does it refer to bins of a discrete retinal space, or to the neighbors? Only in the method section it then becomes clear that the authors mean apparently the neighbors. Such fundamental points should be however make clear already when discussing Eq. 1 in the main text.

- The notation of the equation is strange and I am not sure what " $(p_i | V_i)^{\text{right}}$ " is supposed to indicate. Is this supposed to be some sort of a probability? Is this a function of V_i ? In this case it should be $p(V_i)$.

- In general eq. 1 is hard to understand without further explanation. As this is a probability of right turn during a bout it should be in the range $[0,1]$ but in principle a sum of $w_i \cdot (p_i | V_i)$ could yield numbers outside of this range if not additional constraints on them are enforced. Also the last term "-0.5" just appears without any explanation at all. I assume its related to symmetry but in particular to a non expert reader, it would be completely opaque. It is not even clear whether the term -0.5 is outside the of second sum (i.e. being "applied" only once to the rhs of eq. 1) or is it within the sum. Here brackets would be helpful to avoid misinterpretation.

- Regarding the weights w_i there are some expressions given in the Method sections (lines 786-689) also referring to Fig 2E and Fig 3D, but the relation between the expressions and the

experiments results is unclear.

- Also it is unclear how the bias ($p|V$) is extracted from the experimental data. The authors refer to the figures for showing the $p(\text{turn right})$ versus time but this may show quite some variation over time during the application of the stimulus. So is the bias an average over time, or the maximum value or something else?

4) A minor comment: The caption of Supplementary Figure S1 does not seem to match the panels starting from panel C.

Reviewer #2:

Remarks to the Author:

In their manuscript "Precise visuomotor transformations underlying collective behavior in larval zebrafish", Harpaz and colleagues present an elegant behavioural and modelling exploration of "social" interactions in larval zebrafish of three different ages. They show that across the three ages, left/right turning behaviour in the presence of simple dot stimuli and/or conspecifics can be accurately understood in a relatively straightforward model that takes into account a small number of key variables in stimulus parameters, perhaps most notably the vertical size. The model performs well also in explaining responses to bilateral conflict (i.e. 2 stimuli, one presented to each eye).

In hand, the authors show some interesting age dependent effects, for example that younger animals tend to show weak repulsion while older animals switch to relatively strong attraction (presumably leading to shoaling). This also includes the first (to my knowledge) demonstration that 7 dpf larval zebrafish do in fact "avoid" each other a bit.

Overall, there is a lot to like. The experiments are generally well designed and executed and explained in an accessible manner. Moreover, I particularly like the demonstration of the vertical over horizontal stimulus bias, and even more the visuo-ecologically sensible explanation that this might be linked to improved signal invariance vertically as this does not depend on fish angle. This result, in itself, is a nice place to start from when trying to understand the actual circuits of the retina, presumably in future work. More generally, the results as described are convincing. Accordingly, I support publication of this work in principle. However, I do have a few thoughts that may be useful.

1. "Clutter". The authors use this collective term quite freely to denote a variety of properties of the stimulus, for example its size. They also show nicely that it matters little if you show a single long vertical stimulus, or three dots that approximately cover the same visual space. But then, surely, the three dots "clutter" the visual field more than the single elongated stimulus? There is more to look at. More edges, more variation in contrast over visual space etc. In my mind, clutter is linked to the entropy in the stimulus, but the authors do not look at entropy (which would be quite easy to do). Entropy would also be quite high in nature at times, for example if a fish swims past some vegetation. In contrast, what the authors present is dots on a uniform background. Is clutter really the best term to use? I found it quite confusing.

2. Contrast/Polarity. All stimuli tested were dark, on a bright background (unless I misunderstood). This is reasonable because presumably fish look "dark" to each other in most cases. However it does make one wonder, would the algorithm also work with inverted stimulus contrast? I am not suggesting this as a new experiment, which would be less natural than what the authors have done, but I should at least like to see some discussion/acknowledgment that the stimulus space explored only one half in terms of possibilities regarding polarity. One reason that this matters is in view of eventually understanding retinal circuits that supply these central decision circuits which come in On and Off flavours that differ from each other in many more ways than mere polarity (for example at the level of temporal response profiles or wavelength dependence)

3. Have the authors considered changing sizes of retinal neurons and thus of retinal receptive

fields as the animals grow? By and large, younger animals have relatively larger receptive fields (in angular size), because retinal neurons do not grow in size on par with eye-growth. Could that by itself explain some of the behavioural switches seen as the animal grows? This should also be considered in the context of the animals reevaluating what is and isn't a dangerous/attractive stimulus. For example, 7 dpf fish will eat anything that is $\sim < 5^\circ$ visual angle (which is presumably also why they don't swim away from these stimuli). But a 21 dpf zebrafish will presumably quite happily try to eat a $> 5^\circ$ target. It would be good to see some consideration around these kinds of "competing" reasons for the observed behavioural changes, perhaps as part of the discussion.

4. While I really like the idea of vertical over horizontal size measurements for looking at conspecifics, I do wonder what this means for vegetation. Most plants will also be long, vertical and dark stimuli. Would the fish then not turn away from vegetation, possibly leaving themselves exposed to predators in the open water? It would be good to hear the authors' thoughts on this possibility, perhaps in the discussion.

Signed: Tom Baden

Reviewer #3:

Remarks to the Author:

This is a well-written manuscript and a great example of the utility of being able to manipulate a system to characterize its computations vs deriving the computations from observations of unperturbed output measures. Experiments and analyses are well described and follow a logical flow. I especially appreciate the explicit and testable predictions from the modeling component of the study and the use of the model to reproduce the experimental results from behavior observations that led to less precise insights when compared to the VR experiments. The experimental findings and the elegant demonstration of the utility of experimental manipulations and computational modeling will be of interest to a wide audience.

I really only have a few minor comments and suggested changes for the authors.

Minor changes:

In my view, the term "retinal clutter" is not a good representation of your results. The use of clutter makes me imagine a much less discerning or precise computation. I think your findings that fish integrate over the vertical dimension and average the resulting values over the horizontal dimension is exciting and is not well represented by "clutter". As a tongue-in-cheek example, if I had a stack of books cluttering up my desk I wouldn't call my desk less cluttered just because my colleague placed a smaller stack of books next to it. But that is the equivalent of what your results show. "Averaged integration of vertical clutter" would be a mouthful, but you hopefully get my point.

Lines 267-268: The values you give here ($< 9^\circ$ for 14 dpf and $< 40^\circ$ for 21 dpf) don't correspond to the figure.

Line 278: "in both age groups (Fig. 3C, left)". Only data for 21 dpf are shown.

Line 375: I suggest you change "Constraining" to "Predicting"

Some editing for typos is needed:

I am not sure if "over development" is correct, please check (e.g. line 19)

Check for proper usage of that vs which and associated commas (restrictive vs non-restrictive clauses): e.g. line 24 "retinal clutter, that"

Generally, check for punctuation.

I noticed some instances of plural/singular mix-up. For example:

line 73 "elicit social response"

Figure 1 legend : "and dispersions also decreases"

Point by point response to reviewers

Dear Madam/Sir

We have now generated a point-by-point response to all of the reviewer's questions and concerns (below). Briefly the substantial changes to the manuscript are:

1. We have added new data and experiments to the paper, represented as two new supplementary figures. These figures and data are meant to address two of the main concerns raised by the reviewers. The first is regarding the specific responses of the fish to the motion of the stimuli (reviewer 1), and the second is regarding the generalizability of our findings to different stimulus types (reviewer 2).
2. We have significantly revised the mathematical description of the models both in the main text and in the methods, and added new sections to the methods, clearly describing our quantification of fish responses to the VR stimuli.
3. We significantly revised the text in the discussion, results, figure legends and methods to address the specific concerns and suggestions raised by all reviewers.

Again we would like to thank the reviewers and editors for their thoughtful comments and effort, we believe the manuscript has significantly improved due to these revisions.

All the best

Roy and Florian

REVIEWER COMMENTS

Reviewer #1 (Remarks to the Author):

In this work the authors investigate what features of the visual input affect certain aspect of collective behavior (dispersion versus cohesion). The research is primarily experimental but contains also a modeling component. The research addresses a timely and novel questions with some state of the art methods like the usage of virtual reality essays in animal behavior: How does collective behavior in fish emerge from sensory inputs? How do individuals integrate visual inputs from their neighbors and respond to it with movement?

The paper is well written and comprehensible (for most part, see detailed comment below), the figures are are of good quality. The results are novel and interesting, in fact also quite surprising like the role of the vertical extension of the visual projection. Overall, this result are of potential interest to scientists working on experiments and modeling of collective animal behavior and neuroscience, as well neighboring disciplines, and may thus may have some broader impact. Thus overall I think the paper may be potentially suited for Nat Comms. However, I see some issues that need to be addressed first:

1) If I understand the authors correctly, their result suggest that there is difference in response between moving or non-moving stimuli. This would indicate that optical flow does not seem to play a role for the turning response, which indeed appears surprising and should be discussed in more detail.

We assume that the reviewer intended to say "... their results suggest that there is **no** difference in response between moving and non-moving stimuli...".

We thank the reviewer for this question. It was not our intention to claim that motion has no role in turning responses but rather that retinal occupancy, without any motion is sufficient to elicit strong repulsive (7 dpf) or repulsive and attractive (14, 21 dpf) behaviors. Following the reviewer's question we conducted additional experiments that tested the specific role of motion in our assay: First, we asked whether a moving stimulus could be more powerful than a stationary one, which would be in line with many aspects of visual psychophysics and saliency of stimuli induced by motion. Indeed, we found that when we directly compared stationary against moving stimuli, fish responses were stronger to moving objects compared to stationary ones, regardless of motion direction. We report these findings as additional supplementary material to the manuscript.

Second, we asked whether the direction of motion might also bias the turning response. This is motivated by assays such as the optomotor response, where animals are known to bias their turning to follow the direction of whole field motion. Specifically, we generated stimuli where object position and the direction of motion are congruent in terms of the expected turning direction (e.g. object on the left moving in a clockwise direction- both position and motion should elicit a rightward turn), and compared these responses to stimuli when they are incongruent (e.g. object on the left moving in CCW direction). We found that indeed, congruent stimuli

elicited slightly stronger responses compared to incongruent ones. However, the magnitude of this difference is very small, which is probably due to the fact that only a small portion of the visual field is being stimulated in these assays. We have summarized these results, together with the motion effects, in additional supplementary figure panels (see also inserted below) and explain them now in the text.

Left: Probability to turn right per bout when a moving dot of 9° is presented to the left eye of a 7 dpf fish, moving either in clockwise or counterclockwise direction, and a stationary dot of the same size is presented to the right eye. Bold lines represent mean over fish, and shaded areas are SEM (N=32 fish). Right: Probability to turn right per bout averaged over the entire stimuli duration. Bars represent means and error bars represent SEM, same data as shown on the left. Moving stimuli elicit a stronger response than stationary ones, yet motion direction has only a negligible effect.

2) There seems to be inaccuracies in the terms used in the quantification of the continuous time stochastic process. In figures throughout the paper, the authors show curves for probability for turning right versus time. This is however ill defined. In continuous time stochastic processes a probability has to be defined with respect to a time interval, i.e. it is a product of a probability rate and a time interval. So either the authors implicitly binned time (e.g. according to the frame rate of recordings), which should be clearly stated, they mean in fact probability rates (which is unlikely given the numerical values) or something is odd here.

Of course one could consider individual bouts, and then for each bout (a discrete event) define a probability to turn right. In this case the probability rate would be then set by in the probability rate of single bouts, but this does not seem to be what is meant here.

We thank the reviewer for pointing our attention to the fact that the quantification of fish responses to the visual stimuli are not clear enough. We have revised our wording in the figure legends and we have added a new expanded section in the methods explaining in more detail our calculations. In short, we indeed mean the probability to turn right per bout, which is calculated as the fraction of right turns out of all turns (right + left turns) within a specified time bin (and hence the probability to turn left is simply 1-the probability to turn right). This is equivalent to the probability of the fish to turn right given a bout at any moment in time, and our treatment is justified by the fact that the bout rate of the fish is constant throughout the experiments (Fig. S3I). As such, we can be certain that it is only the bias in direction that is influenced by the stimuli. We hope these changes clarify the issue. We'd be happy to change the nomenclature of our plots, axes and legends according to the recommendations of the reviewer if they feel it is still confusing to the reader.

3) the model description and parametrization in the main text and in particular in the methods is too short, lacking some important information, which make it hard to really understand what the model does, in particular for non experts:

Again we thank the reviewer for this helpful comment. We have revised and expanded our description of the models and revised the mathematical equations, the notations and the accompanying explanations. Again we hope this makes the modeling part clear and accessible to the readers. See specific responses to each of the comments below:

- From just reading the section containing eq. 1 in the main text its not clear what the index i is? Does it refer to bins of a discrete retinal space, or to the neighbors? Only in the method section it then becomes clear that the authors mean apparently the neighbors. Such fundamental points should be however make clear already when discussing Eq. 1 in the main text.

We thank the reviewer for pointing out this inconsistency. We meant to describe the process as relying entirely on global retinal occupancy, in which classification of neighbors is unneeded, following our experimental findings and also reflected in our conceptual circuit model. We revised the main text and methods parts to emphasize this point. In the actual implementation of the models we used a simplifying assumption, where we neglected occlusions among fish, as we found that such a simplification does not affect the average group properties over an entire simulated experiment in the groups sizes used here. Given this simplification, we treated the average vertical occupancy generated by a neighbor simply as its projected vertical dimension on the retina as the two are equivalent. These simplifications were used for run-time speed purposes only, since implementing ray casting (as done in the group swimming experiments) is slow. We now describe these steps and simplifications clearly in the methods part, and of course we include the codes for all models in the Github repository.

- The notation of the equation is strange and I am not sure what " $p_i | V_i$ " is supposed to indicate. Is this supposed to be some sort of a probability? Is this a function of V_i ? In this case it should be $p(V_i)$.

The reviewer is correct - we revised all mathematical notations in the main text and in the methods. Indeed the correct form is $p(\text{turn right} | V_i^{\text{right/left}})$ - the probability to turn right given an occupied visual sub angle of size v_i on the right/left visual fields.

- In general eq. 1 is hard to understand without further explanation. As this is a probability of right turn during a bout it should be in the range [0,1] but in principle a sum of $w_i \cdot p_i | V_i$ could yield numbers outside of this range if not additional constraints on them are enforced. Also the last term "-0.5" just appears without any explanation at all. I assume its related to symmetry but in particular to a non expert reader, it would be completely opaque. It is not even clear whether the term -0.5 is outside the of second sum (i.e. being "applied" only once to the rhs of eq. 1) or is it within the sum. Here brackets would be helpful to avoid misinterpretation.

Again, we agree with the reviewer and apologize for the confusion. We revised the way equation 1 is presented, yet holding the same calculations true and we believe it is clearer now. We attach the text explaining eq. 1 from the manuscript:

"Specifically, each occupied horizontal visual sub-angle θ_i on the retina, elicits a turning bias based on its integrated vertical size v_i : $\text{bias}(v_i) = p(\text{turn right} | v_i) - 0.5$ (positive values represent a rightward bias, negative is leftward), where the age relevant turn probabilities $p(\text{turn right} | v_i)$ were learned from VR experiments (Fig. 2C, 3A-B, right). Next, in accordance with the monocular and binocular integration algorithms observed in VR, these turning biases are averaged over all occupied visual angles θ on each side of the fish, and finally the (signed) average responses are linearly added, such that:

$$p(\text{turn right} | \theta^{\text{left}}, \theta^{\text{right}}) = \left[\sum_i^{\theta^{\text{left}}} w_i^{\text{left}} \cdot \text{bias}(v_i^{\text{left}}) + \sum_i^{\theta^{\text{right}}} w_i^{\text{right}} \cdot \text{bias}(v_i^{\text{right}}) \right] + 0.5$$

Where w_i is the relative weight assigned to each response bias and represents either a weighted average $w_i = v_i / \sum v_i$ (7 dpf) or a simple average $w_i = 1/N_\theta$ (14 and 21 dpf) of the monocular turning biases (Fig. 4A). The intercept 0.5 centers the summed responses around that value and $p(\text{turn right} | \theta^{\text{left}}, \theta^{\text{right}})$ is bounded between 0 and 1 using a piecewise linear function (see Methods). "

Regarding the constraints of the equations, the weights w_i average the responses within each eye in accordance with the observed averaging of responses to multiple stimuli presented to a single eye (Fig. 2E, S4C-D, 3D). For binocular integration we used a linear summation of the averaged monocular biases, based on our experimental findings showing that binocular turning

responses (in all ages) can be described by a linear sum of the single eye biases (Fig. 2F, S4E, 3E), and we center it around 0.5 using an intercept (see equation). This sum can indeed extend beyond 0 and 1, and we therefore use a piecewise linear function (shown below in blue) to constrain its values to that range. We now state this clearly in the text describing equation 1 and in the methods section explaining binocular integration. Such a representation is akin to a biological ceiling effect, in which a stimulus presented to one eye might be so strong that the addition of a second stimulus to the other eye would not cause a linear increase in the response, if it is already saturated. We also consider an alternative representation using a logistic function of the form:

$p(\text{turn right} | v^{\text{left}}, v^{\text{right}}) = 1. / (1 + e^{-a[\text{bias}(v^{\text{left}}) + \text{bias}(v^{\text{right}})])}$ that can also naturally map to a response probability between 0 and 1 (shown below in red).

We chose the piecewise linear function here, since all the binocular responses in our experiments (Fig. 2F, 3E and S4E) are well approximated by a linear addition of the monocular response biases and therefore captured well by the central (linear) component of either function.

- Regarding the weights w_i there are some expressions given in the Method sections (lines 786-689) also referring to Fig 2E and Fig 3D, but the relation between the expressions and the experiments results is unclear.

We have revised the text explaining the models in the methods section (specifically parts d and e), also incorporating the additional comments raised by the reviewer in this section. We hope that the new version clears the uncertainty pointed to by the reviewer.

Specifically, our experimental findings show that fish responses to the combined presentation of monocular stimuli were best described by a weighted average of the response biases recorded for each stimulus presented alone at 7 dpf (Fig. 2E, S4C-D), and by a simple average of the biases at ages 14 and 21 dpf (Fig. 3D). Therefore the weights in the age relevant models follow these findings. Again, we now state this clearly in the main text and in the methods.

- Also it is unclear how the bias ($p|V$) is extracted from the experimental data. The authors refer to the figures for showing the $p(\text{turn right})$ versus time but this may show quite some variation over time during the application of the stimulus. So is the bias an average over time, or the maximum value or something else?

Indeed the probability to turn right for a given vertical size is calculated as the average of the time binned probabilities over the trial duration. We now make this more explicit in the figure legends for figures 2C (right) and 3A,B (right).

4) A minor comment: The caption of Supplementary Figure S1 does not seem to match the panels starting from panel C.

Thank you for pointing this out. We corrected the panel letters, and double checked all our figure panels again.

Reviewer #2 (Remarks to the Author):

In their manuscript "Precise visuomotor transformations underlying collective behavior in larval zebrafish", Harpaz and colleagues present an elegant behavioural and modelling exploration of "social" interactions in larval zebrafish of three different ages. They show that across the three ages, left/right turning behaviour in the presence of simple dot stimuli and/or conspecifics can be accurately understood in a relatively straightforward model that takes into account a small number of key variables in stimulus parameters, perhaps most notably the vertical size. The model performs well also in explaining responses to bilateral conflict (i.e. 2 stimuli, one presented to each eye).

In hand, the authors show some interesting age dependent effects, for example that younger animals tend to show weak repulsion while older animals switch to relatively strong attraction (presumably leading to shoaling). This also includes the first (to my knowledge) demonstration that 7 dpf larval zebrafish do in fact "avoid" each other a bit.

Overall, there is a lot to like. The experiments are generally well designed and executed and explained in an accessible manner. Moreover, I particularly like the demonstration of the vertical over horizontal stimulus bias, and even more the visuo-ecologically sensible explanation that this might be linked to improved signal invariance vertically as this does not depend on fish angle. This result, in itself, is a nice place to start from when trying to understand the actual circuits of the retina, presumably in future work. More generally, the results as described are convincing. Accordingly, I support publication of this work in principle. However, I do have a few thoughts that may be useful.

1. “Clutter”. The authors use this collective term quite freely to denote a variety of properties of the stimulus, for example its size. They also show nicely that it matters little if you show a single long vertical stimulus, or three dots that approximately cover the same visual space. But then, surely, the three dots “clutter” the visual field more than the single elongated stimulus? There is more to look at. More edges, more variation in contrast over visual space etc. In my mind, clutter is linked to the entropy in the stimulus, but the authors do not look at entropy (which would be quite easy to do). Entropy would also be quite high in nature at times, for example if a fish swims past some vegetation. In contrast, what the authors present is dots on a uniform background. Is clutter really the best term to use? I found it quite confusing.

We understand that addressing the global visual occupancy elicited by neighbors as ‘clutter’ might cause confusion, and might be ill defined. We have therefore revised our definitions throughout the paper, replacing visual clutter with retinal occupancy as an ‘umbrella term’ which we define here not as a scalar, but rather as a matrix, where each element is either 1 (black, occupied) or 0 (white, not occupied). Horizontal averaging and vertical integration are then simple mathematical operations on the rows and columns of this matrix, respectively.

We acknowledge that this is a strong simplification of the true stimulus space, and that we ignore details such as shades of gray, spatial and temporal filtering, and many other retinal features that are clearly present already in our young animals (as correctly pointed out by the reviewer below with respect to ‘On’ and ‘Off’ channels for example). However, we believe that our success in predicting fish behavior simply based on this reduced feature space, makes a strong statement about the explanatory power of our reduced model; i.e. it allows us to speculate which retinal information streams specifically feed into the underlying circuit, and which probably do not. We now introduce the term ‘Retinal Occupancy’ and its definition explicitly in the text.

This definition also allows one to get a straightforward estimate of the entropy, which is indeed an interesting and intriguing concept to consider. However, our data show that larval zebrafish appear to integrate primarily over occupancy rather than measure entropy, since adding additional edges into our stimulus does not appear to change its strength (three dots vs ellipse). Thus these young animals seem to prioritize occupancy vs edges. We specifically mention this distinction in the text (lines 238 - 240).

2. Contrast/Polarity. All stimuli tested were dark, on a bright background (unless I misunderstood). This is reasonable because presumably fish look “dark” to each other in most cases. However it does make one wonder, would the algorithm also work with inverted stimulus contrast? I am not suggesting this as a new experiment, which would be less natural than what the authors have done, but I should at least like to see some discussion/acknowledgment that the stimulus space explored only one half in terms of possibilities regarding polarity. One reason that this matters is in view of eventually understanding retinal circuits that supply these central decision circuits which come in On and Off flavours that differ from each other in many more

ways than mere polarity (for example at the level of temporal response profiles or wavelength dependence)

The reviewer is making an excellent point. We now added experiments testing responses to light-on-dark stimuli (i.e. inverted polarity) and we found that they elicit qualitatively similar responses but their efficiency is slightly smaller. We have now added a supplementary Figure that illustrates this effect (see also below) and describe it in the results section. We also discuss the ecological significance of darker objects over a light background as well as the constraints it allows us to put on our retinal circuit model (lines 399 - 402).

3. Have the authors considered changing sizes of retinal neurons and thus of retinal receptive fields as the animals grow? By and large, younger animals have relatively larger receptive fields (in angular size), because retinal neurons do not grow in size on par with eye-growth. Could that by itself explain some of the behavioural switches seen as the animal grows? This should also be considered in the context of the animals reevaluating what is and isn't a dangerous/attractive stimulus. For example, 7 dpf fish will eat anything that is $\sim 5^\circ$ visual angle (which is presumably also why they don't swim away from these stimuli). But a 21 dpf zebrafish will presumably quite happily try to eat a $>5^\circ$ degree target. It would be good to see some consideration around these kinds of "competing" reasons for the observed behavioural changes, perhaps as part of the discussion.

The relation between the ontogeny of collective behavior and the development of the eyes is intriguing. It is indeed possible that attraction is inhibited in young larvae due to a poor capability to distinguish small objects (e.g. $<5^\circ$), even if the neural circuits to perform such behavior are in place. We can, however, assert that young larvae are able to distinguish between larger stimuli (e.g. the graded repulsion from 9° - 36°) and that these same size stimuli begin to elicit attraction only at older ages. Therefore, we believe that a weaker resolving power at a young age (7 dpf) might indeed contribute to the lack of attraction at this age, but we expect that the maturation of additional mechanisms (other than improved visual capabilities) plays a role in the emergence

of attraction as fish grow. We now address these different possibilities explicitly in the discussion.

4. While I really like the idea of vertical over horizontal size measurements for looking at conspecifics, I do wonder what this means for vegetation. Most plants will also be long, vertical and dark stimuli. Would the fish then not turn away from vegetation, possibly leaving themselves exposed to predators in the open water? It would be good to hear the authors thoughts on this possibility, perhaps in the discussion.

The reviewer raises an excellent point, and indeed stretching the object over the entire retina (as in the case of vegetation) might elicit a different class of responses. We therefore hypothesize that the extension of vertical stimuli beyond a certain vertical size, will cause non-monotonic changes in the turning responses of the fish, perhaps inhibiting repulsive turning responses altogether. Neuronal types that can contribute to such non monotonic responses have been described previously in the mammalian cortex, such as hyper-complex and end-stopped neurons, and it is not known if such neurons exist in the zebrafish. We now discuss these points in the discussion.

Signed: Tom Baden

Reviewer #3 (Remarks to the Author):

This is a well-written manuscript and a great example of the utility of being able to manipulate a system to characterize its computations vs deriving the computations from observations of unperturbed output measures. Experiments and analyses are well described and follow a logical flow. I especially appreciate the explicit and testable predictions from the modeling component of the study and the use of the model to reproduce the experimental results from behavior observations that led to less precise insights when compared to the VR experiments. The experimental findings and the elegant demonstration of the utility of experimental manipulations and computational modeling will be of interest to a wide audience.

I really only have a few minor comments and suggested changes for the authors.

Minor changes:

In my view, the term “retinal clutter” is not a good representation of your results. The use of clutter makes me imagine a much less discerning or precise computation. I think your findings that fish integrate over the vertical dimension and average the resulting values over the horizontal dimension is exciting and is not well represented by “clutter”. As a tongue-in-cheek example, if I had a stack of books cluttering up my desk I wouldn’t call my desk less cluttered

just because my colleague placed a smaller stack of books next to it. But that is the equivalent of what your results show. “Averaged integration of vertical clutter” would be a mouthful, but you hopefully get my point.

We thank the reviewer for this comment and their office desk example, which clearly addresses the problems in simply using ‘clutter’ to describe the visual aspects of our stimuli. We agree, and it appears that other members of our review panel seem to share this opinion :-). As described above (see response to reviewer 2), we have now revised our terminology throughout the paper substituting clutter with ‘retinal occupancy’ and ‘averaged retinal occupancy’ or ‘integrated retinal occupancy’ where appropriate. We also explicitly introduce and define the term ‘retinal occupancy’ not as a scalar in this context, but rather as a matrix where each element is either 1 (black, occupied) or 0 (white, not occupied). We hope these terms are now much clearer.

Lines 267-268: The values you give here ($<9^\circ$ for 14 dpf and $<40^\circ$ for 21 dpf) don’t correspond to the figure.

We thank the reviewer for pointing our attention to this discrepancy which we have corrected.

Line 278: “in both age groups (Fig. 3C, left)”. Only data for 21 dpf are shown.

We added a reference to supplementary figure S5C showing similar results for 14 dpf larvae.

Line 375: I suggest you change “Constraining” to “Predicting”

We agree that constraining is not the best usage here. We now call this section “A conceptual model of the underlying neurobiological circuits”, which we hope better serves the purpose of this section.

Some editing for typos is needed:

I am not sure if “over development” is correct, please check (e.g. line 19)

We are also not sure :-) We checked with various native speakers, and they seemed to agree that “over development” is OK. However, maybe the editorial team can also weigh in here.

Check for proper usage of that vs which and associated commas (restrictive vs non-restrictive clauses): e.g. line 24 “retinal clutter, that”

We thank the reviewer for this comment, hopefully we were able to correct all places in the text where such errors occurred.

Generally, check for punctuation.

I noticed some instances of plural/singular mix-up. For example:

line 73 “elicit social response”

Figure 1 legend : “and dispersions also decreases”

Again we thank the reviewer for helping us correct our typos :-) We have re-checked and tried to find them all.

Reviewers' Comments:

Reviewer #1:

Remarks to the Author:

The authors have significantly revised the manuscript and clarified/addressed the comments/issues raised in my initial report in a satisfactory manner.

The only issue I found is that the github repo:

<https://github.com/harpazone/Modeling-larvae-social-behavior>
is not accessible. I assume the repo is still set to private.

Once this is resolved, I endorse the acceptance of the manuscript with Nature Communications based on the interesting results.

Reviewer #2:

Remarks to the Author:

The authors have done a good job at addressing the majority of my comments, and I am happy to for this to go ahead in principle, perhaps pending some mini updates as per the below that could be checked at the editorial level going forward.

Specifically, about my previous point 3 (eye/retina size). I did not really mean whether or not the fish can resolve the stimulus (I suspect they can at either age, certainly their cone-spacing would suffice as seen e.g. in Zimmermann 2018 Curr Biol). It was more about whether or not the very same retinal circuits (RGC level, presumably) will inevitably become more small-field tuned with age because their dendritic trees do not grow on par with radial expansion of the eye. E.g. a hypothetical "Type 1 RGC" might have a 20 degree receptive field at 7 dpf, but a 5 degree receptive field in adults. In this way, the exact same wiring in the eye, using the same types of cells, could lead to a behavioural size-selective switch simply because things grow in 3D, not 2D

Point by point response to reviewers - second revision

We include below a response to the two additional issues raised by the reviewers.

All the best

Roy and Florian

REVIEWER COMMENTS

Reviewer #1 (Remarks to the Author):

The authors have significantly revised the manuscript and clarified/addressed the comments/issues raised in my initial report in a satisfactory manner.

The only issue I found is that the github repo:

<https://github.com/harpazone/Modeling-larvae-social-behavior>

is not accessible. I assume the repo is still set to private.

Once this is resolved, I endorse the acceptance of the manuscript with Nature Communications based on the interesting results.

We thank the reviewer for her/his support of our study and apologize for not being able to access the repository. We have corrected the issue and the repository is now readily accessible.

Reviewer #2 (Remarks to the Author):

The authors have done a good job at addressing the majority of my comments, and I am happy to for this to go ahead in principle, perhaps pending some mini updates as per the below that could be checked at the editorial level going forward.

Specifically, about my previous point 3 (eye/retina size). I did not really mean whether or not the fish can resolve the stimulus (I suspect they can at either age, certainly their cone-spacing would suffice as seen e.g. in Zimmermann 2018 Curr Biol). It was more about whether or not the very same retinal circuits (RGC level, presumably) will inevitably become more small-field tuned with age because their dendritic trees do not grow on par with radial expansion of the eye. E.g. a hypothetical "Type 1 RGC" might have a 20 degree receptive field at 7 dpf, but a 5 degree

receptive field in adults. In this way, the exact same wiring in the eye, using the same types of cells, could lead to a behavioural size-selective switch simply because things grow in 3D, not 2D

We thank the reviewer for his support of our study and for the additional clarification of his original point. We understand that RGCs in the retina can become more highly tuned to smaller size objects due to developmental changes in the growing eye. Yet, we do not see how this change, on its own, will also lead to a behavioral switch from repulsion to attraction. Perhaps a more straightforward prediction is that older larvae would therefore repel away from even smaller objects, since their RFs are now smaller, yet this is not what we observed. In addition, at 21 dpf, larvae turn towards a wide range of stimulus sizes, even as large as 45° (a retinal size that elicits very strong repulsion at younger ages). It is hard for us to speculate how the mere change in RF size can account for this complete switch in behavior, without some rewiring.

We therefore address the reviewer's comment in a more general form in the discussion where we comment on the developmental changes of the eye.

Specifically, we now say:

“ Attraction to low visual occupancy was observed only in older larvae. We hypothesize that the neural circuits that support attraction behavior develop with age. However, the lack of attraction to smaller angular sizes at 7 dpf, can also stem from a limitation of the developing visual system, where retinal ganglion cell receptive field size decreases with age (72). Still, the increased tendency to repel from objects of increasing angular sizes at 7 dpf and the tendency of older larvae to attract to these same angular sizes, supports the notion of a developmental ‘switch’ in the tendency to attract to neighbors, that cannot simply be explained by a developmental change in size tuning of retinal receptive fields. “